# The Roles of the miRNAome and Transcriptome in the Ovine Ovary Reveal Poor Efficiency in Juvenile Superovulation

**DOI:** 10.3390/ani11010239

**Published:** 2021-01-19

**Authors:** Xiaosheng Zhang, Chunxiao Dong, Jing Yang, Yihai Li, Jing Feng, Biao Wang, Jinlong Zhang, Xiaofei Guo

**Affiliations:** Institute of Animal Husbandry and Veterinary Medicine, Tianjin Academy of Agricultural Sciences, Tianjin 300381, China; zhangxs0221@126.com (X.Z.); dongchx16@163.com (C.D.); YangJing9599@163.com (J.Y.); sheepteam@163.com (Y.L.); fengjing86114@163.com (J.F.); wb18536655745@163.com (B.W.)

**Keywords:** miRNAome, transcriptome, juvenile superovulation, JIVET, ovary, sheep

## Abstract

**Simple Summary:**

Using the technology of juvenile superovulation, more follicles can be acquired in juvenile animals than in adult animals. However, oocytes derived from the follicles of juvenile animals are usually of poor quality, meaning that they have lower levels of subsequent maturation and embryonic development. In the present study, we used an exogenous hormone treatment to stimulate Hu sheep in order to compare the differences in ovarian superovulation effects and serum hormone secretion in juvenile and adult sheep. Differentially expressed microRNA (miRNA) and messenger RNA (mRNA) from the ovaries of juvenile and adult Hu sheep were then investigated using high-throughput sequencing technology to reveal the formation mechanism of large numbers of follicles and poor oocyte quality in juvenile ovaries under superovulation treatment. We found that molecules of oar-miR-143 and follicle-stimulating hormone receptor (*FSHR*), among others, might regulate follicle formation, while oar-miR-485-3p, oar-miR-377-3p, and pentraxin 3 (*PTX3*), among others, may be associated with oocyte quality. The results will help us to identify miRNAs and mRNAs that could be used to predict ovarian superovulation potential and oocyte quality in the future.

**Abstract:**

Juvenile superovulation can provide a wealth of oocyte material for embryo production, animal cloning, and genetic modification research, but embryos derived from juvenile oocytes show poor efficiency in subsequent developmental capacity. In order to reveal the formation mechanism of large numbers of follicles and poor oocyte quality in juvenile ovaries under superovulation treatment, differentially expressed microRNAs (miRNAs) and messenger RNAs (mRNAs) were characterized and investigated in the ovaries of lambs and adult sheep using high-throughput sequencing technology. The majority of differentially expressed miRNAs (337/358) were upregulated in lamb libraries. The expression levels of mRNAs related to hormone receptors (follicle-stimulating hormone receptor, *FSHR*; luteinizing hormone/choriogonadotropin receptor, *LHCGR*; estrogen receptor 1, *ESR1*), steroid hormone secretion (cytochrome P450 family 11 subfamily A member 1, *CYP11A1*; cytochrome P450 family 17 subfamily A member 1, *CYP17A1*; cytochrome P450 family 19 subfamily A member 1, *CYP19A1*), and oocyte quality (pentraxin 3, *PTX3*; BCL2 apoptosis regulator, *BCL2*; caspase 3, *CASP3*) were significantly different between the lamb and adult libraries. The miRNA aor-miR-143, which targets *FSHR*, was highly and differentially expressed, and *PTX3* was predicted to be targeted by oar-miR-485-3p and oar-miR-377-3p in the ovine ovary. A considerable number of miRNAs were predicted to inhibit *ESR1* expression in lamb ovaries. In conclusion, oar-miR-143 and *FSHR* molecules, among others, might regulate follicle formation, and oar-miR-485-3p, oar-miR-377-3p, and *PTX3*, among others, may be associated with oocyte quality. These identified miRNAs and mRNAs will be beneficial for the prediction of ovarian superovulation potential and screening of oocytes.

## 1. Introduction

In the breeding of domestic animals (sheep, cattle, pigs, etc.), limiting factors such as the reproductive cycle and generation interval affect their genetic progress. Exploring the reproductive potential of juvenile animals can provide an exciting shortcut to solve the above-mentioned limiting factors [1,2]. Although the gonadal systems of prepubertal females are not fully developed, primordial germ cells are already present in their ovaries [3]. Under the stimulation of superovulation hormones, prepubertal females of a suitable age can produce considerably more antral follicles than adult females [1]. However, for unknown physiological reasons, these antral follicles from prepubertal females cannot ovulate normally, and follicular aspiration on the juvenile ovary is necessary for oocyte collection [4]. The oocytes are then matured and fertilized in vitro, and the produced embryos are transplanted into the recipients. Compared with the technique of multiple ovulation and embryo transfer (MOET) applied in adult females, the above process, so-called juvenile in vitro embryo transfer (JIVET), is not smooth, and it has been found that embryos derived from juvenile oocytes show lower rates of development [5]. We suspected that the level of ovarian development in juvenile animals may be related to miRNA and mRNA expression levels, which contribute to limit its application on a large scale.

The miRNAs are a class of highly conserved small non-coding RNAs that can modulate mRNA and protein expression [6]. It has been proven that miRNAs participate in various physiological processes, including hormone secretion [7], organogenesis [8], cell proliferation [9], apoptosis [10], differentiation [11], metabolism [12], and reproduction control [13]. It has been estimated that miRNAs, which occupy a very small proportion of predicted genomes, could regulate up to one-third of all protein-coding genes in higher eukaryotes [14]. In recent years, many studies have further shown that miRNAs are involved in mammalian follicular, oocyte, and luteal development [15,16]. Otsuka et al. proved that miR-17-5p and let-7b contribute to the infertility of mutant mice [17]. Thereafter, the role of miRNAs in animal ovaries has been an area of constant interest for researchers. Yin et al. suggested that in mouse granulosa cells, miRNAs of miR-320 target E2F1 and SF-1, which can regulate proliferation and steroid production in follicular development, while miR-320 expression is regulated by follicle-stimulating hormone (FSH) secretion; moreover, miR-383 enhances miR-320-mediated suppression of granulosa cell proliferation [18]. With the advancement of high-throughput sequencing technology, more miRNAs in ovaries have been identified and correlated with reproduction-related traits [19,20,21]. 

Many studies have shown that oocytes derived from juvenile animals show poor subsequent development [22]. In 2015, our team also reported that the low global DNA methylation and hydroxymethylation in oocytes derived from lambs are associated with less subsequent developmental potential [23]. Recently, ovine mRNA expression in prepubertal and adult superstimulated follicle granulosa cells was studied using RNA sequencing technology, and more than 300 differentially expressed mRNAs were reported; in particular, the beta-estradiol upstream regulator in the prepubertal ovary was of interest [24]. In the present study, we used an exogenous hormone treatment to stimulate Hu sheep in order to compare the differences in ovarian superovulation effects and serum hormone secretion in juvenile and adult sheep; then, differentially expressed miRNAs and mRNAs from the ovaries of juvenile and adult Hu sheep were investigated using high-throughput sequencing technology to reveal the formation mechanism of large numbers of follicles and poor oocyte quality in juvenile ovaries under superovulation treatment. The results will help us to identify miRNAs and mRNAs that might be used to predict ovarian superovulation potential and oocyte quality in the future.

## 2. Materials and Methods 

### 2.1. Animals and Superovulation Treatment 

The two groups comprised six Hu sheep each. Animals were aged 1 month (lamb group) or 24 months (adult group), and were selected for superovulation treatment in the spring. The lambs were given intramuscular injections of FSH (Sansheng, Ningbo, China) every 12 h for 2 days (250 IU in total); pregnant mare serum gonadotropin (PMSG, Sansheng, Ningbo, China) was administered intramuscularly at the last FSH treatment (250 IU in total). An intravaginal controlled progesterone release device (CIDR; Pharmacia and Upjohn Co., Hartwell, Australia) insert was placed inside each adult sheep on Day 0, and removed on Day 12. FSH (300 IU in total) was injected intramuscularly every 12 h for the last 2 days. PMSG (360 IU in total) was injected intramuscularly 6 h after the CIDR was removed. All experimental procedures were approved by the Science Research Department of the Tianjin Academy of Agricultural Sciences (TAAS; Tianjin, China). Ethical approval for animal surgery was authorized by the animal welfare committee of TAAS (No. 2020009). 

### 2.2. Sample Collection and RNA Extraction

Jugular venous blood and ovaries of lambs were obtained 14 h after PMSG injection, and jugular venous blood and ovaries of adult sheep were obtained 36 h after CIDR removal. Samples of 10 mL jugular blood were collected in BD SST Tubes (BD, Franklin Lakes, NJ, USA) and held at room temperature for 30 min, and then centrifuged at 1500× *g* for 10 min. The product of serum was used for determination of FSH, luteinizing hormone (LH), progesterone (P_4_), and estradiol (E_2_) concentrations. Each hormone was measured using an iodine [^125^I] radioimmunoassay kit (BNIBT, Beijing, China) according to the instructions. Three sheep were randomly selected from each group for anesthesia; following anesthesia, the abdominal skin and muscle were cut open with a scalpel to a length of 5cm, and the left ovary was pulled out of the body. After tubal ligation, the left ovary was removed and frozen instantly in liquid nitrogen. In the liquid nitrogen freezing environment, each whole ovary was ground into powder. Finally, the powder was collected for RNA extraction using the Trizol method. 

### 2.3. Library Construction and Sequencing

For miRNA library construction, small RNAs in the size range of 18–30 nt were isolated from total RNA based on 15% denaturing polyacrylamide gel electrophoresis (PAGE). Subsequently, the 3′ adapters were added and 36–44 nt RNAs were enriched. The 5′ adapters were then also ligated to the RNAs. The ligation products were reverse-transcribed by PCR amplification, and the PCR products with a size of 140–160bp were enriched to generate a cDNA library. The libraries were sequenced through an Illumina HiSeqTM 2500 at Gene Denovo Biotechnology Co. (Guangzhou, China).

For mRNA library construction, poly-T oligo-attached magnetic beads were used to purify the RNA samples, and mRNAs were subsequently fragmented. The fragments were then reversed for cDNA synthesis using random primers. With the action of polymerase I and RNase H, second-strand cDNAs were synthesized. Next, a QiaQuick PCR extraction kit (Qiagen, Venlo, Netherlands) was used to purify the cDNA fragments. The cDNA fragments underwent end repair with poly (A) added and were ligated to Illumina sequencing adapters. The second-strand cDNA was then digested and selected by agarose gel electrophoresis. After PCR amplification, the libraries were sequenced by Gene Denovo Biotechnology Co. (Guangzhou, China) using the Illumina HiSeqTM 4000 platform.

### 2.4. Read Filtering and Differential Expression Analysis

For miRNA libraries, contaminants with an adaptor, poly A, and low-quality reads, or those shorter than 18nt, were filtered to generate clean reads. In order to remove rRNAs, tRNAs, snRNAs, and snoRNAs, we aligned the clean reads in GenBank and Rfam11.0 [25]. Clean reads were also aligned with the miRNA precursors/mature miRNAs of all animals in miRBase 22.0 (http://www.mirbase.org/), and the sequence and count of miRNAs in each library were displayed. The prefix “oar-miR” indicates that the miRNA has been included in the miRBase database for the species Ovis aries; no prefix indicates that the miRNA has not been included in the species database. Moreover, “miR-number-x” and “miR-number-y” indicate that the sequence can be aligned to -5 p and -3 p for the corresponding miRNA in miRBase for other animal species. Principal component analysis (PCA) was performed based on the sample correlation matrix. Differentially expressed miRNAs in lamb and adult groups were screened with a threshold of false discovery rate (FDR) < 0.05, log_2_ (fc) > 1 or log_2_ (fc) < –1.

For mRNA libraries, contaminants with an adapter, poly-N, and low-quality reads were removed to generate clean reads. Based on Hisat2 v2.1.0 (https://daehwankimlab.github.io/hisat2/) [26], the clean reads were mapped into the ovine genome (Oar_v4.0). The read number of each mRNA was calculated using featureCounts v1.5.0-p3 (https://sourceforge.net/projects/subread/files/subread-1.5.0-p3/) [27]. PCA was again performed based on the sample correlation matrix. Differentially expressed mRNAs were screened with a threshold of FDR < 0.05, log_2_ (fc) > 1 or log_2_(fc) < –1.

### 2.5. Targeted Relationship between Differentially Expressed miRNAs and mRNAs

The 2–8 nt sequences, starting from the 5′ end of miRNA, were chosen as seed sequences for prediction with the 3′-UTR of target mRNAs. RNAhybrid (version 2.1.2) [28], svm_light (version 6.01) [29], Miranda (version 3.3a) [30], and TargetScan (version 7.0) [31] were used to predict miRNA targets. The intersections of these results were more likely to be chosen as predicted miRNA target genes. Upon combining the mRNAs targeted by differentially expressed miRNAs and the differentially expressed mRNAs in the present study, an intersection was generated. We then constructed an miRNA–mRNA expression regulatory network based on the negative correlation between the expression of miRNAs and mRNAs using Cytoscape_v3.7.1 [32].

### 2.6. Enrichment Analysis by GO and KEGG

Predicted target mRNAs of differentially expressed miRNAs and differentially expressed mRNAs were used separately for Gene Ontology (GO) and Kyoto Encyclopedia of Genes and Genomes (KEGG) enrichment analysis. These mRNAs were blasted to all known animals in the GO and KEGG databases. Gene numbers for terms in GO and KEGG were calculated, and a hyper-geometric test was used to find significantly enriched GO terms and KEGG pathways. A term or pathway with a corrected p-value (Q-value) < 0.05 was defined as significantly enriched.

### 2.7. Real-Time PCR Validation 

Real-time PCR was performed on the miRNAs of oar-miR-143, miR-224-x, oar-miR-377-3p, oar-miR-485-3p, and oar-miR-487a-3p, as well as mRNAs of the follicle stimulating hormone receptor (*FSHR*), luteinizing hormone/choriogonadotropin receptor (*LHCGR*), estrogen receptor 1 (*ESR1*), pentraxin 3 (*PTX3*), and the BCL2 apoptosis regulator (*BCL2*). The primers of these molecules are shown in Appendix A. Total RNA from sequencing samples was separately reversed into cDNA for miRNA and mRNA validation assay using the miScript Reverse Transcription Kit (Qiagen, Dusseldorf, Germany) and Prime Script^®^ RT reagent Kit (Takara Bio Inc., Dalian, China). All cDNAs from miRNAs and mRNAs were separately mixed to generate standard samples for miRNA and mRNA validation assay. U6 and RPL19 were separately used as reference miRNA and mRNA to calculate the relative expression level with the 2^−ΔΔCT^ method. The real-time PCR procedure for miRNA was as follows: incubation at 95 °C for 10 min, followed by 40 cycles at 94 °C for 15 s and 60 °C for 45 s. The real-time PCR procedure for mRNAs was as follows: incubation at 95 °C for 5 min, followed by 40 cycles at 95 °C for 10 s and 60 °C for 30 s. All reactions were performed in triplicate. 

### 2.8. Statistical Analysis

The chi-square test was used to analyze categorical variables of follicle numbers. Duncan’s multiple range test program in ANOVA was adopted to analyze continuous variables of hormone concentration and mRNA/miRNA expression based on the SAS 8.0 software (SAS Institute Inc., Cary, NC, USA). All of the results are presented as mean ± SE. To assess the relationship between high-throughput sequencing and real-time PCR in mRNA/miRNA expression assay, the Spearman correlation was calculated based on SPSS version 20.0 (SPSS Inc., Chicago, IL, USA)). 

## 3. Results

### 3.1. Measurement of Follicle Numbers and Serum Hormone in Lamb and Adult Sheep under Superovulation Treatment

Under stimulation with the exogenous hormone, superovulation was successfully performed in lambs and adult Hu sheep. The follicle numbers and serum reproduction hormone levels are presented in Table 1. The mean super-stimulated follicle number (total of two ovaries) in lambs was 70.17 ± 5.14, which was much higher than that in adult sheep (20.17 ± 2.65) with a high significance (*p* < 0.01). However, the concentrations of FSH, LH, progesterone (P_4_), and estradiol (E_2_) in jugular vein serum all showed no significant difference between the lamb and adult groups after superovulation treatment. 

### 3.2. Overview of Sequencing Data

Three biological repetitions in each group (lamb and adult) were selected to construct small RNA and mRNA libraries for sequencing analysis. For the six small RNA libraries, a total of 98,479,314 high-quality reads were obtained, and a total of 94,250,039 clean tags (Appendix A), with lengths ranging from 18 to 30 nt (Figure 1A), were generated after discarding contaminants. As shown in Appendix A, average conserved miRNAs (including exist_mirna, exist_mirna_edit and known_mirna) accounted for 89.59% ± 1.60% of the total clean tags in the lamb libraries, which was significantly lower than 96.29 ± 0.40% in the adult libraries (*p* < 0.05). This means that more miRNAs may be found in prepubertal ovine ovaries in the future. 

For the six mRNA libraries, an average of 77,998,161 clean reads in each library were generated after discarding adapter–adapter and low-quality reads (Appendix A). The Q20 values of the mRNA libraries were all higher than 97%. Appendix A shows the results of clean reads mapping the ovine genome. More than 96% and 93% of clean reads were totally mapped and uniquely mapped, respectively, in the ovine genome for all six mRNA libraries. In summary, the data quality of the miRNA and mRNA libraries in the present study was high enough for them to be used for subsequent statistical analyses.

### 3.3. Differentially Expressed miRNA and mRNA in the Lamb and Adult Libraries

A total of 1819 miRNAs were detected in all six miRNA libraries, and random clustering of the replicates in the PCA of all detected miRNAs showed that the first principal component explained 70.6% of the total variation between lamb and adult groups (Figure 1B). As shown in Figure 1C, 358 miRNAs were differentially expressed between the lamb and adult libraries (FDR < 0.05, log_2_ (fc) > 1 or log_2_ (fc) < –1) (Appendix A). Compared to the adult libraries, 337 miRNAs were upregulated in the lamb libraries, which accounted for the majority of differentially expressed miRNAs, and only 21 miRNAs were down regulated. Heat-map analysis of differentially expressed miRNAs showed a separation between the individuals of lamb and adult sheep (Figure 1D). The top 10 highly and differentially expressed miRNAs that might play important roles in sexual maturation of the ovary are listed in Table 2.

A read count greater than 1 defined detectable genes in mRNA expression level, and a total of 17,842 mRNAs were detected in the six mRNA libraries (Appendix A). Correlation coefficients within the lamb group were all higher than 0.979, and correlation coefficients within the adult group were all higher than 0.969 (Figure 2A). PCA revealed a clear distinction in mRNA expression of ovaries between lamb and adult individuals (Figure 2B). Finally, 3150 mRNAs were found to be differentially expressed in the two groups (Figure 2C, Appendix A). Compared to the adult libraries, 1359 and 1791 mRNAs were upregulated and downregulated, respectively, in the lamb libraries. In order to examine the differences in the sensitivity of juvenile and adult sheep to superovulation hormones, as well as expression levels of key mRNAs related to oocyte quality, we selected some mRNAs to emphasize their expression differences between the two groups. As shown in Table 3, the expression levels of hormone receptor genes (follicle-stimulating hormone receptor, *FSHR*; luteinizing hormone/choriogonadotropin receptor, *LHCGR*; estrogen receptor 1, *ESR1*), steroid hormone secretion genes (cytochrome P450 family 11 subfamily A member 1, *CYP11A1*; cytochrome P450 family 17 subfamily A member 1, *CYP17A1*; cytochrome P450 family 19 subfamily A member 1, *CYP19A1*; steroidogenic acute regulatory protein, *STAR*; 3 beta-hydroxysteroid dehydrogenase, *HSD3B*; 17-beta-hydroxysteroid dehydrogenase type 1, *HSD17B1*), and genes related to oocyte quality (pentraxin 3, *PTX3*; growth differentiation factor 9, *GDF9*; DNA methyltransferase 1, *DNMT1*, caspase 3, *CASP3*; insulin like growth factor binding protein 4, *IGFBP4*; BCL2 apoptosis regulator, *BCL2*; apoptosis regulator BAX, *BAX*) were significantly different in the ovaries of lambs and adult sheep (FDR < 0.05). Heat map analysis of differentially expressed mRNAs also showed a separation between the individual lambs and adult sheep (Figure 2D).

### 3.4. Analysis of the Targeted Relationship between Differentially Expressed miRNAs and mRNAs

Based on pairing between the 5′ end of miRNAs and the 3′ untranslated region of target mRNAs, miRNAs can modulate mRNA expression at the post-transcriptional level [6]. Taking the intersection of the differentially expressed miRNA target mRNAs with differentially expressed mRNAs, and screening with a negative correlation between the expression of the miRNA and mRNA, we constructed an miRNA–mRNA expression regulatory network. As shown in Appendix A, 2194 mRNAs were predicted to be targeted by the 358 differentially expressed miRNAs. The differentially expressed miRNAs that targeted key genes (*FSHR*, *LHCGR*, *CYP11A1*, *STAR*, *PTX3*, and *ESR1*) are listed in Table 4 and Figure 3. The miRNA oar-miR-143, which targeted *FSHR* at a high level and with differential expression in ovaries between lamb and adult sheep [33,34], and the *PTX3* gene, which was related to oocyte quality and targeted by oar-miR-377-3p, oar-miR-485-3p, oar-miR-487a-3p, miR-496-x, and miR-767-x, are particularly noteworthy. The most notable feature was the part of the miRNA–mRNA network related to the *ESR1* and *PTX3* gene (Figure 3). 

### 3.5. GO and KEGG Enrichment Analysis for Target Genes of Differentially Expressed miRNAs

For target genes of miRNAs that were differentially expressed in the lamb and adult groups, a total of 21,194 target genes were mapped to GO in terms of the cellular component (CC): 19,819 to molecular function (MF) and 21,888 to biological processes (BP) (Appendix A). Comparing the reference gene background in CC, MF, and BP, a total of 105, 99, and 458 GO terms were significantly enriched (Q-value < 0.05), respectively. GO terms of the hormone-mediated signaling pathway (GO: 0009755), cellular response to hormone stimulus (GO: 0032870), response to hormone (GO: 0009725), histone methylation (GO: 0016571), protein methylation (GO: 0006479), and histone lysine methylation (GO: 0034968) were significantly enriched in the target genes (Q-value < 0.05). This suggests that the ovaries of lambs and adult sheep differ in their response to hormones and in their methylation levels. In addition, GO terms of epithelial cell proliferation (GO: 0050673), cell differentiation (GO: 0030154), programmed cell death (GO: 0012501), and regulation of MAPK cascade (GO: 0043408) were significantly enriched (Q-value < 0.05), which suggests that cell proliferation, differentiation, and apoptosis behaviors in the ovaries of lambs and adult sheep are also different.

KEGG pathway enrichment showed that a total of 16,627 background genes were annotated in 344 biological functions. In the present study, we found that 13,290 target genes for differentially expressed miRNAs were annotated in KEGG pathways. A total of 180 pathways were significantly enriched (Q-value < 0.05) (Appendix A). It is worth mentioning that progesterone-mediated oocyte maturation (ko04914), oocyte meiosis (ko04114), the estrogen signaling pathway (ko04915), the TGF-beta signaling pathway (ko04350), and the MAPK signaling pathway (ko04010) were all significantly enriched in target genes for miRNAs differentially expressed in the lamb and adult groups (Q-value < 0.05). These results suggest that oocytes in lambs and adults have different maturation levels.

### 3.6. GO and KEGG Enrichment Analysis of Differentially Expressed mRNAs

A total of 1944, 1763, and 2004 differentially expressed mRNAs were annotated in the GO terms of BP, CC, and MF, respectively. Compared to the reference gene background in CC, MF, and BP, a total of 39, 11, and 309 GO terms were significantly enriched (Q-value < 0.05), respectively (Appendix A). In particular, the GO terms of cell proliferation (GO: 0008283), steroid metabolic process (GO: 0008202), sterol metabolic process (GO: 0016125), steroid biosynthetic process (GO: 0006694), regulation of secretion (GO: 0051046), and MAPK cascade (GO: 0000165) were significantly enriched (Q-value < 0.05). This means that the significantly enriched GO terms in mRNA and miRNA analysis had similar results in terms of cell proliferation and hormone secretion regulation.

Subsequently, enrichment analysis of the KEGG pathway was performed based on the differentially expressed mRNAs. A total of 1442 mRNAs were mapped to the relevant KEGG pathways, and 48 KEGG pathways were shown to be significantly enriched (Q-value < 0.05) (Appendix A). In the analysis of mRNA expression level, the pathways of glutathione metabolism (ko00480) and ovarian steroidogenesis (ko04913) attracted our research attention. However, neither of these two significant pathways presented as enriched in the miRNA KEGG analysis. In addition, many significant enrichment pathways in the miRNA analysis were also absent from the mRNA KEGG analysis. 

### 3.7. Real-Time PCR Validation of mRNA and miRNA Associated with Hormone Receptor and Oocyte Quality

The expression levels of five miRNAs (oar-miR-143, miR-224-x, oar-miR-377-3p, oar-miR-485-3p, and oar-miR-487a-3p) and five mRNAs (*FSHR*, *LHCGR*, *PTX3*, *BCL2*, and *CASP3*) were validated in the ovaries of lambs and adult sheep using real-time PCR (Figure 4). In these selected molecules, *FSHR* was targeted by oar-miR-143 [34,36], and *PTX3* was targeted by miR-224-x, oar-miR-377-3p, oar- oar-miR-485-3p, and miR-487a-3p [37]. mRNA and miRNA expression values from each sample, detected by high-throughput sequencing and real-time PCR were used for Pearson correlation analysis (Appendix A). The correlation coefficients of the two methods in expression detection of the 10 selected molecules ranged from 0.635 to 0.978, which demonstrates that the expression levels detected by high-throughput sequencing were reliable.

## 4. Discussion

JIVET technology can be used to shorten the generation interval and accelerate the progress of genetic selection, and it can also provide a wealth of materials for research on embryonic stem cells, animal cloning, and genetic modification [3,22,38]. Nevertheless, embryos derived from JIVET exhibit a significantly lower survival rate [38]. The maturation status of the cumulus cell [39], oocyte cytoplasm [40], oocyte nuclear [41], and follicular fluid microenvironment in juvenile ovaries [42,43,44] have been suggested to explain the poor efficiency of JIVET. In the present study, we focused on the initial stage of JIVET: ovaries under the treatment of superovulation; therefore, the miRNAome and transcriptome were compared between juvenile and adult ovaries to identify the mechanism of inferior oocyte capture by JIVET. 

We initially identified 358 miRNAs that were differentially expressed in the lamb and adult libraries, and the majority of these miRNAs were upregulated in the lamb libraries. Considering that miRNAs can negatively regulate gene expression at the post-transcriptional level, we speculate that much more gene expression in lamb ovaries was suppressed. Under superstimulatory treatment of prepubertal and adult sheep, Wu et al. found 175 downregulated genes vs. 136 upregulated genes in prepubertal granulosa cells using RNA sequencing [24], and they also found 155 downregulated proteins vs. 88 upregulated proteins in prepubertal follicular fluid using iTRAQ [43]. In our results, we also found 1791 downregulated mRNAs vs. 1359 upregulated mRNAs in the ovaries of lamb. The results from the research of Wu et al. and our mRNA results verify the findings of the miRNAs comparisons. Based on an enzyme immunoassay evaluating the effects of 80 miRNAs on human granulosa cells, Sirotkin et al. showed that miR-145 enhanced progesterone production up to 5-fold, and miR-144 reduced estradiol secretion up to 4.5-fold [45]. In this study, miR-145-x and miR-144-y were also highly expressed and were down- and upregulated, respectively, in the lamb libraries. In addition, downregulation of the miRNAs oar-miR-10a and miR-145-x in lambs was speculated in the present study, implicating suppression of granulosa cell proliferation by the targeting of brain-derived neurotropic factor (*BDNF*) and activin receptor IB (*ACVRIB*) [7,46,47]. 

The cyclic activity of the ovary is mainly influenced by reproductive hormones [48,49]; under the stimulation of FSH and LH, the follicles undergo primary, secondary, and antral stages, and only a few of them reach the pre-ovulatory stage [50]. After superovulation treatment in the present study, we found that there was no significant difference in the concentrations of FSH and LH in jugular vein serum between lambs and adult sheep. However, the exciting finding was that *FSHR* and *LHCGR* expression levels in lamb ovaries were significantly higher than those in adult sheep ovaries. Specifically, at the same hormone level, lambs have an increased sensitivity to hormones due to the increased expression of hormone receptors in ovarian tissue. Based on RT-PCR, immunohistochemistry, and Western blot results, Scarlet et al. also showed that *FSHR* was present in the ovaries of prepubertal fillies, and found that *FSHR* was abundant in the ovarian stroma cells of neonate fillies but not of adults [51]. *LHCGR* is also considered to be an important target gene induced by FSH and is closely related to steroid hormone metabolism and ovarian follicular development [52]. Recently, Michalovic et al. reported that cumulus–oocyte complexes (CoCs) and the granulosa cells (GCs) collected from Holstein heifers that matured under FSH treatment presented increased mRNA levels of genes promoting follicular growth (*FSHR*) and estradiol synthesis (*CYP19A1*) [53]. In this study, the mRNA expression levels of the steroid hormone or estradiol synthesis genes (*CYP11A1*, *CYP17A1*, *CYP19A1*, *STAR*, *HSD3B*, and *HSD17B1*) in lamb ovaries were also significantly higher than those in adult sheep, which means that estradiol synthesis is occurring. However, we found that mRNA expression of the *ESR1* gene in lambs ovaries was significantly inhibited, which may be related to the physiological state of prepuberty and detrimental to follicular development. In mice, females lacking the *ESR1* gene are infertile and non-receptive to males [54]. In the present study, a considerable number of miRNAs were predicted to target the *ESR1* mRNA in lamb ovaries, and these miRNAs were significantly highly expressed in lamb. Couse and Korach demonstrated that *ESR1* plays an important role in follicular growth and development and subsequent ovulation [55]. Liu et al. showed that the estradiol (E2) estrogen receptors (ERs) system in follicular granulosa cells plays a dominant role in controlling oocyte meiotic resumption in mammals [56]. In human and mouse cumulus cells, downregulation of *ESR1* could be related to oocyte competence and is likely to be the driver of the expression changes highlighted in the PI3K/AKT pathway [57].

The poor efficiency of juvenile superovulation is mainly reflected in the poor quality of oocytes collected from prepubertal female animals [58]. Zhang et al. reported that the *PTX3* gene in human cumulus cells is highly associated with oocyte development [59]. In the present study, the downregulated *PTX3* mRNA levels in lamb ovaries suggested that oocytes derived from lambs may be of poor quality. However, growth differentiation factor-9 (*GDF-9*), a gene upstream of *PTX3*, was upregulated in lambs. Kona et al. reported that the expression of GDF9 was the highest in ovine primordial follicles compared with the preantral, early antral, antral, and large antral stages [60]. Upregulated *GDF9* in the lamb group indicated that ovaries from lambs contain more primordial follicles than adult sheep. However, we found that mRNA expression levels of the *BCL2* gene in lamb ovaries were downregulated, but we did not find a difference in *BAX*. It is well known that the expression ratio of Bax and Bcl-2 determines the propensity for cell apoptosis or survival [61]. In the present study, because the ratio of Bax and Bcl-2 was significantly lower in the lamb group than the adult group, we speculated that the proliferative capacity of granulosa cells in lamb ovaries is not good. *DNMT1* is responsible for maintaining methylation patterns throughout DNA replication, and we found that its mRNA expression is higher in lamb ovaries. However, Fang et al. found that CoCs derived from lamb matured in vitro with a lower mRNA expression of *DNMT1* [23]. This suggests that CoCs and ovarian tissue of lamb may be inconsistent in methylation levels. *CASP3* involved in apoptosis was proven to be responsible for follicular atresia [62]; the higher level of mRNA expression of *CASP3* in lamb ovaries in this study may reflect that a more follicular atresia events occur. IGFBP4 mRNA transcripts were detected in human follicular fluid, and this has been demonstrated to be associated with oocyte maturation and embryo development [63]. This suggests that the lower mRNA expression level of *IGFBP4* detected in lamb ovaries may lead to the emergence of low-quality oocytes.

miRNA and mRNA together have been proven to play important roles in the process of follicular development and oocyte maturation in the ovary [50,64,65,66,67,68]. Using the combined miRNAome and transcriptome results, we constructed miRNA regulatory networks for key genes in the ovine ovary. What needs to be emphasized is that downregulation of *ESR1* and *PTX3* may be attributed to the targeted inhibition of a considerable number of miRNAs in the lamb ovary. Unfortunately, we found that miR-224, which targets the *PTX3* gene in sow ovaries [37], was not negatively correlated with the expression level of *PTX3* in this study. The miRNAs oar-miR-485-3p, oar-miR-487a-3p, and oar-miR-377-3p, which are species-specific to sheep and, target *PTX3* in the ovine ovary, should be further studied in the future. Last year, Zhang et al. demonstrated using bovine granulosa cells that overexpression of miR-143 led to decreased secretion of estradiol, and silencing of miR-143 was associated with increased estradiol production [34]. This inhibitory action of miR-143 on estradiol synthesis is mediated via its target gene, *FSHR* [7,34]. A previous study also reported high expression of miR-143 in granulosa cells of primary, secondary, and antral follicles associated with a negative regulation of FSH-induced estradiol production [36]. In the present study, the downregulation of oar-miR-143 in lamb ovaries may have directly contributed to the upregulated of *FSHR* expression. The expression relationship between miR-513-x and *LHCGR* confirmed the results in human granulosa cells [35].

Each miRNA might target hundreds of genes, and in turn, each of these genes might be regulated by several miRNAs [69]. Therefore, placing too much emphasis on the expression levels of single miRNAs or mRNAs is not advisable [50]. In order to comprehensively analyze the differentially expressed miRNAs and mRNAs in lamb and adult sheep ovaries, GO term and KEGG pathway enrichment analysis was performed. Compared with the enrichment analysis of differentially expressed mRNAs, many more GO terms and KEGG pathways were enriched in the target genes of differentially expressed miRNAs. However, mRNA and miRNA target genes were all enriched in GO terms of cell proliferation, which is consistent with the results reported on the protein level [43,70]. However, we found little difference in KEGG enrichment between differentially expressed mRNA and target genes of miRNA. This result may be attributed to the difference in spatiotemporal expression between miRNA and mRNA, as miRNA needs to find the 3′ untranslated regions of target mRNA before it can play a regulatory role [6]. On the basis of our observations of combination of the mRNAs and target genes of miRNAs in KEGG analysis, we firmly believe that there are some significant differences between the ovaries of lambs and adult sheep in terms of cell proliferation, hormones response, and oocyte development level.

## 5. Conclusions

In conclusion, the majority of differentially expressed miRNAs (337/358) were upregulated in the lamb libraries. The expression levels of mRNAs related to hormone receptors (*FSHR*, *LHCGR*, and *ESR1*), steroid hormone secretion (*CYP11A1*, *CYP17A1*, and *CYP19A1*), and oocyte quality (*PTX3*, *BCL2*, and *CASP3*) showed a significant difference between the lamb and adult sheep libraries. The miRNA oar-miR-143, which targets FSHR, was highly differentially expressed, and *PTX3* was predicted to be targeted by oar-miR-485-3p and oar-miR-377-3p in the ovine ovary in this study. A considerable number of miRNAs were predicted to inhibit *ESR1* expression in lamb ovaries. We suppose that molecules of oar-miR-143 and *FSHR*, among others, might regulate follicle formation, and oar-miR-485-3p, oar-miR-377-3p, and *PTX3*, among others, may be associated with oocyte quality. These identified miRNAs and mRNAs will be beneficial for potential ovarian superovulation prediction and oocyte screening.

## Figures and Tables

**Figure 1 animals-11-00239-f001:**
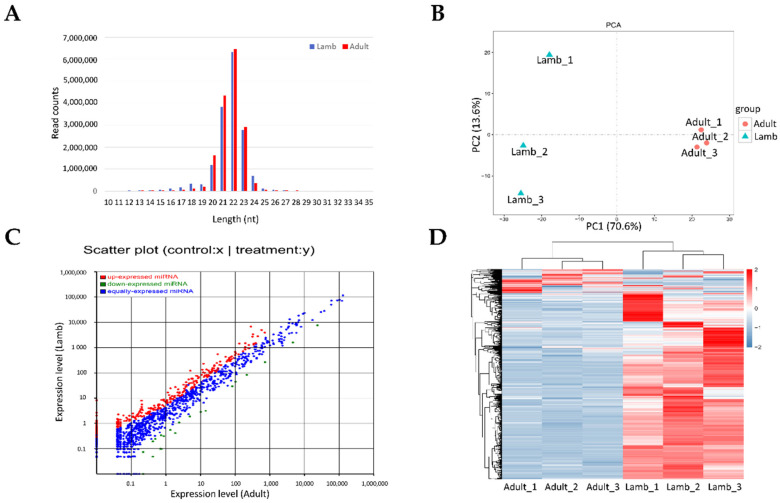
miRNA expression analysis of the ovaries of lamb and adult sheep under superovulation treatment. (**A**) Sequence length distribution for miRNA sequencing. (**B**) Principal component analysis (PCA) of expression of miRNA values in the lamb and adult sheep libraries. (**C**) Screen of differentially expressed miRNAs. (**D**) Heat map analysis of differentially expressed miRNAs.

**Figure 2 animals-11-00239-f002:**
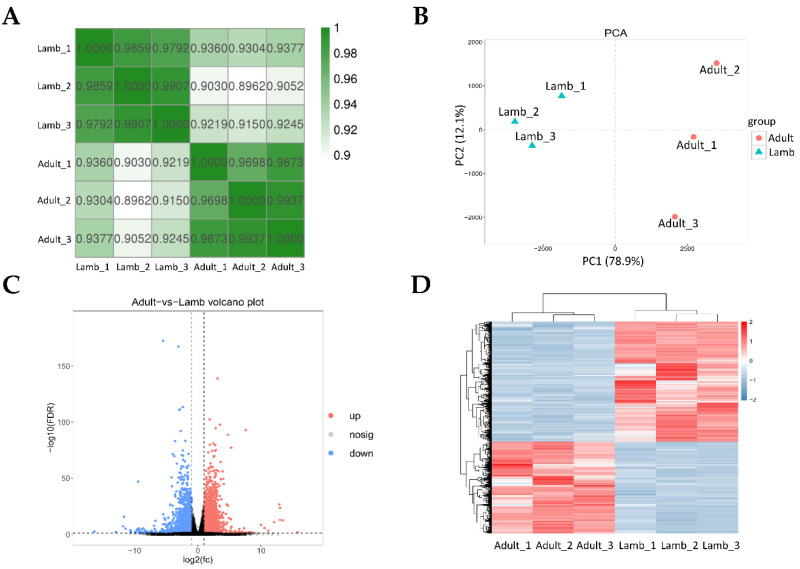
mRNA expression analysis for the ovaries of lambs and adult sheep under superovulation treatment. (**A**) Pearson correlation for mRNA expression values across samples. (**B**) PCA of mRNA expression values of the lamb and adult libraries. (**C**) Volcano plot of distribution for differentially expressed mRNAs. (**D**) Heat map analysis of differentially expressed mRNAs.

**Figure 3 animals-11-00239-f003:**
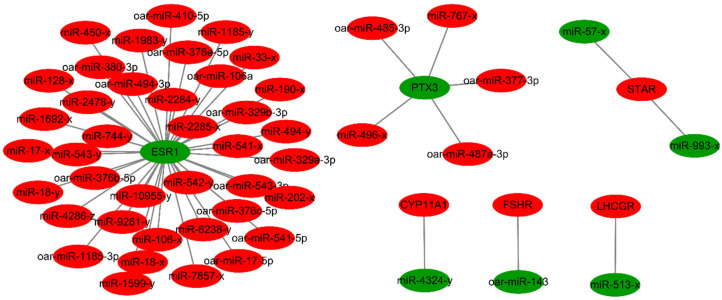
miRNA–mRNA expression regulatory network related to genes associated with the hormone receptor, steroid hormone secretion, and oocyte quality. Molecules in red and green denote upregulation and downregulation in the lamb libraries, respectively.

**Figure 4 animals-11-00239-f004:**
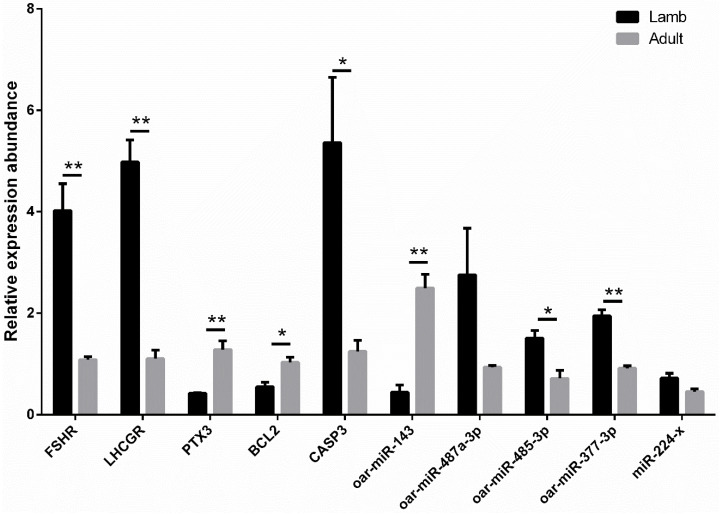
Real-time PCR validation of mRNA associated with hormone receptor and oocyte quality. * Significant difference at *p* < 0.05, ** significant difference at *p* < 0.01.

**Table 1 animals-11-00239-t001:** Measurement of follicle numbers and serum hormone levels in lamb and adult sheep.

Group	Number of Animals	Average Number of Follicles	FSH (mIU/mL)	LH (mIU/mL)	P_4_ (10^−2^ ng/mL)	E_2_ (pg/mL)
Lamb	6	70.17 ± 5.14 ^a^	3.65 ± 0.35	3.66 ± 0.37	1.36 ± 0.24	4.43 ± 0.37
Adult	6	20.17 ± 2.65 ^b^	3.61 ± 0.32	3.88 ± 0.10	1.33 ± 0.11	4.67 ± 0.30

Note: Values with different superscript letters in the same column represent highly significant difference (*p* < 0.01). Values with no superscript letters in the same column have no significant difference (*p* > 0.05).

**Table 2 animals-11-00239-t002:** Top 10 highly and differentially expressed miRNAs in the lamb and adult sheep libraries.

miRNA	Lamb_Mean	Adult_Mean	log_2_(fc)	*p*-Value	FDR
miR-145-x	7649.92	23730.58	–1.63	3.43 × 10^−25^	1.04 × 10^−22^
miR-144-y	6678.34	286.18	4.54	3.79 × 10^−9^	9.32 × 10^−8^
miR-451-x	4928.68	578.77	3.09	4.28 × 10^−4^	3.33 × 10^−3^
miR-450-x	3914.41	721.04	2.44	6.42 × 10^−6^	9.05 × 10^−5^
miR-486-x	3379.14	327.35	3.37	1.22 × 10^−4^	1.11 × 10^−3^
miR-424-x	2801.34	448.96	2.64	3.14 × 10^−7^	5.34 × 10^−6^
miR-202-x	2696.87	316.41	3.09	1.93 × 10^−11^	7.79 × 10^−10^
oar-miR-10a	1594.46	4637.33	–1.54	2.30 × 10^−11^	9.09 × 10^−10^
oar-miR-3958-3p	1497.39	392.70	1.93	4.71 × 10^−2^	1.65 × 10^−1^
oar-miR-409-3p	1485.38	197.24	2.91	5.88 × 10^−11^	2.02 × 10^−9^

Note: miRNAs with a red background represent upregulation in the lamb libraries compared to the adult sheep libraries (false discovery rate (FDR) < 0.05, log_2_(fc) > 1); miRNAs with a green background represent downregulation in the lamb libraries (FDR < 0.05, log_2_(fc) < –1).

**Table 3 animals-11-00239-t003:** mRNAs associated with hormone receptors, steroid hormone secretion and oocyte quality were differentially expressed in the lamb and adult sheep libraries.

Gene	Lamb_Mean	Adult_Mean	log_2_(fc)	FDR	Gene Classification
*FSHR*	5.93	1.49	1.99	4.63 × 10^−7^	HC
*LHCGR*	84.10	16.31	2.37	1.77 × 10^−29^	HC
*ESR1*	8.89	22.16	−1.32	3.77 × 10^−13^	HC
*CYP11A1*	251.29	41.33	2.60	3.66 × 10^−42^	SHE
*CYP17A1*	74.87	4.64	4.01	1.71 × 10^−15^	SHE
*CYP19A1*	50.23	1.37	5.20	2.46 × 10^−14^	SHE
*STAR*	32.56	11.31	1.53	5.09 × 10^−13^	SHE
*HSD3B*	112.84	39.14	1.53	6.76 × 10^−3^	SHE
*HSD17B1*	87.11	6.59	3.73	1.15 × 10^−7^	SHE
*PTX3*	4.02	20.19	−2.33	2.75 × 10^−40^	ROQ
*GDF9*	4.07	1.33	1.61	6.91 × 10^−4^	ROQ
*DNMT1*	28.33	12.30	1.20	2.21 × 10^−39^	ROQ
*CASP3*	7.19	1.58	2.18	2.75 × 10^−8^	ROQ
*IGFBP4*	44.01	74.65	−0.76	2.71 × 10^−6^	ROQ
*BCL2*	3.28	5.38	−0.71	8.84 × 10^−3^	ROQ
*BAX*	16.84	13.98	0.27	2.60 × 10^−1^	ROQ
*BAX/BCL2*	5.13	2.60	0.98	7.75 × 10^−3^	NA

Note: mRNAs with a red background represent upregulation in the lamb libraries compared to the adult sheep libraries (FDR < 0.05, log2 (fc) > 1); mRNA with green background represent downregulation in lamb libraries (FDR < 0.05, log2 (fc) < – 1). HC—represents the gene classified into the hormone receptor gene; SHE—represents the gene classified into the steroid hormone secretion gene; ROQ—represents the gene is classified into genes related to oocyte quality; NA—represents not applicable.

**Table 4 animals-11-00239-t004:** List of differentially expressed miRNAs targeting key genes.

id	Lamb_Mean	Adult_Mean	log2(fc)	FDR	Target Gene	Target Gene Classification
oar-miR-143 [34]	70,716.65	122,228.40	−0.79	1.98 × 10^−11^	*FSHR*	HC
miR-513-x [35]	1.09	2.41	−1.14	2.96 × 10^−4^	*LHCGR*	HC
miR-4324-y	2.84	10.16	−1.84	1.20 × 10^−16^	*CYP11A1*	SHE
miR-57-x	0.05	0.42	−3.13	1.47 × 10^−3^	*STAR*	SHE
miR-993-x	0.34	0.73	−1.11	7.74 × 10^−2^	*STAR*	SHE
oar-miR-485-3p	60.89	5.71	3.41	4.33 × 10^−10^	*PTX3*	ROQ
oar-miR-487a-3p	55.39	4.55	3.61	6.36 × 10^−11^	*PTX3*	ROQ
oar-miR-377-3p	9.39	1.46	2.69	6.53 × 10^−3^	*PTX3*	ROQ
miR-767-x	3.81	0.43	3.16	1.09 × 10^−2^	*PTX3*	ROQ
miR-496-x	0.51	0.01	5.67	2.67 × 10^−2^	PTX3	ROQ

Note: Molecules with a red background represent upregulation in the lamb libraries compared to the adult sheep libraries (FDR < 0.05, log_2_ (fc) > 1); molecules with a green background represent downregulation in the lamb libraries (FDR < 0.05, log_2_ (fc) < –1). HC—the gene is classified into the hormone receptor gene; SHE—the gene is classified into the steroid hormone secretion gene; ROQ—the gene is classified into genes related to oocyte quality.

## Data Availability

The data presented in this study are available on request from the corresponding author. The data are not publicly available to preserve privacy of the data.

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
