# Peer review of "The Roles of the miRNAome and Transcriptome in the Ovine Ovary Reveal Poor Efficiency in Juvenile Superovulation"

_animals, 2021, doi:10.3390/ani11010239_

Round 1

Reviewer 1 Report

This is s good study but not presented very well.  Authors evaluated the role of miRNAome and Transcriptome in the ovine ovary and the underlying signaling/events involved in poor efficiency in juvenile superovulation. Overall, the study is convincing but there are numerous errors/mistakes as well. I have some comments/suggestions about this study which I would like the authors to address before publication.

Comments:

  1. The title of the paper needs to be re-written. The title is ambiguous and not clear.

One example is “The role of miRNAome and Transcriptome in Ovine Ovary Revealed the Poor Efficiency in Juvenile Superovulation”.

  1. Second, the authors should consider how to write the terminologies properly such as “miRNA” instead of “MiRNA” right in the title. This should be carefully considered. Review the whole manuscript thoroughly as i expect similar mistakes and typos in the whole manuscript.
  2. The paper needs English language corrections. One example of the imbalance and vague sentence is the first sentence in the Summary i.e. “Under the technology of juvenile superovulation, more and smaller follicles can 12 be obtained in ovaries of juvenile animals than in the adult animals”.
  3. Line 20, 21, 23 - It should rephrase as “we found that……. while oar-miR-485-3p
  4. Line 65, 66,27, 68 - Sentence is too long

Line 91, 92 - Sentences need reorganization. E.g., each comprised of six Hu sheep, aged 4-week (Lamb group) and 2-year (Adult group) were selected for superovulation treatment.

It would be better to use the same unit for both groups. E.g. 4 weeks and 104 weeks or, 1 month and 24 months.

  1. Line 93 - “every 12 hours in 2 days”. It should be “Every 12 hour for 2 days”
  2. Line 95 - “The adult sheep were inserted with an intra-vaginally CIDR (Pharmacia and Upjohn Co., 95 NSW, Australia) for 12 days and”

At what rate? or just once?

  1. Line 171, 172 - Grammar and sentence structure needs correction
  2. Line 173 – replace “were presented” to “are presented”.
  3. Figure 1 - Not clear/visible. Also, what do the X and Y-axis show in Fig 1 A?
  4. Line 218, 219 - Not clear
  5. Line 226 - Significantly differ
  6. Figure 2 - Not clear/visible
  7. Line 245 and 247-250 - Needs grammatical correction
  8. Line 262 - Difficult to read. It should be restructured as

…. Groups: a total of 21,194 target genes were mapped to GO in terms of the cellular component; 19,819 to molecular function and 21,888 to biological processes.

  1. LINE 284 - Not readable. It should be “A total of 1,944, 1,763 and 2,004 differentially…….”
  2. Line 343 - “expression in lamb ovaries were highly significantly higher than”. These types of errors are all over the manuscript.

Author Response

Reviewer 1: 

This is a good study but not presented very well. Authors evaluated the role of miRNAome and Transcriptome in the ovine ovary and the underlying signaling/events involved in poor efficiency in juvenile superovulation. Overall, the study is convincing but there are numerous errors/mistakes as well. I have some comments/suggestions about this study which I would like the authors to address before publication.

Comments:

  1. The title of the paper needs to be re-written. The title is ambiguous and not clear. One example is “The role of miRNAome and Transcriptome in Ovine Ovary Revealed the Poor Efficiency in Juvenile Superovulation”.

Response:

Thank you for your advice. The title of “Study on MiRNAome and Transcriptome in Ovine Ovary to Reveal the Mechanism of Poor Efficiency in Juvenile Superovulation” was corrected as “The Role of miRNAome and Transcriptome in Ovine Ovary Reveal Poor Efficiency in Juvenile Superovulation”.

  1. Second, the authors should consider how to write the terminologies properly such as “miRNA” instead of “MiRNA” right in the title. This should be carefully considered. Review the whole manuscript thoroughly as i expect similar mistakes and typos in the whole manuscript.

Response:

Thank you for showing our mistake. All words of “MiRNA” in the manuscript were checked and corrected as “miRNA”.

  1. The paper needs English language corrections. One example of the imbalance and vague sentence is the first sentence in the Summary i.e. “Under the technology of juvenile superovulation, more and smaller follicles can be obtained in ovaries of juvenile animals than in the adult animals”.

Response:

The sentence of “Under the technology of juvenile superovulation, more and smaller follicles can be obtained in ovaries of juvenile animals than in the adult animals.” was corrected as “Using the technology of juvenile superovulation, we can acquire more follicles in juvenile animals than in adult animals.”

  1. Line 20, 21, 23 - It should rephrase as “we found that……. while oar-miR-485-3p

Response:

In line 20-23, “We found……. and oar-miR-485-3p……” was corrected as “We found that……. while oar-miR-485-3p……”

  1. Line 65, 66,27, 68 - Sentence is too long. Line 91, 92 - Sentences need reorganization. E.g., each comprised of six Hu sheep, aged 4-week (Lamb group) and 2-year (Adult group) were selected for superovulation treatment. It would be better to use the same unit for both groups. E.g. 4 weeks and 104 weeks or, 1 month and 24 months.

Response:

(a) “In recent years, many studies have further shown that miRNAs are involved in mammalian follicular, oocyte and luteal development [15,16], ovarian function [17], proliferation and apoptosis of ovarian granulosa cells [18], female fertility and reproduction [19], pituitary gonadotropin (FSH and LH) secretion [20], estrogen action [21], prolificacy [22] and seasonal estrus trait formation [23,24].” was corrected as “In recent years, many studies have further shown that miRNAs are involved in mammalian follicular, oocyte and luteal development [15,16].”

 (b) “Each group involved six Hu sheep of 4-week-old (Lamb group) and 2-year-old (Adult group) were selected for superovulation treatment.” was corrected as “The groups comprised 6 Hu sheep aged 1 month (Lamb group) or 24 month (Adult group) selected for superovulation treatment.”

  1. Line 93 - “every 12 hours in 2 days”. It should be “Every 12 hour for 2 days”

Response:

“every 12 hours in 2 days” was corrected as “every 12 hour for 2 days”

  1. Line 95 - “The adult sheep were inserted with an intra-vaginally CIDR (Pharmacia and Upjohn Co., 95 NSW, Australia) for 12 days and”. At what rate? or just once?

Response:

The rate is once. CIDR is a controlled progesterone release device which can be inserted into the ovine vagina on Day 0, and the CIDR was removed on Day 12.

To avoid ambiguity, we modified it in the manuscript as follow: An intravaginal CIDR (controlled progesterone release device; Pharmaciaand Upjohn Co., NSW, Australia) insert was placed in each adult sheep on day 0, and removed on day 12.

  1. Line 171, 172 - Grammar and sentence structure needs correction

Response:

“Under the stimulation of exogenous hormone, six lamb and six adult Hu sheep were successfully performed the treatment of superovulation.” was corrected as “Under the stimulation of exogenous hormone, superovulation was successfully performed in the lamb and adult Hu sheep.”

  1. Line 173 – replace “were presented” to “are presented”.

Response:

“were presented” was corrected as “are presented”

  1. Figure 1 - Not clear/visible. Also, what do the X and Y-axis show in Fig 1 A?

Response:

After modification, Figure 1 has been enlarged, and the titles of the X and Y-axis in Figure 1 have also been added.

  1. Line 218, 219 - Not clear

Response:

“A total of 21,352 genes were detected in all of the mRNA libraries, the correlation coefficients among libraries in each group of lamb and adult were all higher than 0.96” was corrected as “A read count greater than 1 defined detectable genes in mRNA expression level, and a total of 17,842 mRNAs were detected in the six mRNA libraries (Supplementary File 6). Correlation coefficients within the lamb group were all higher than 0.979, and correlation coefficients within the adult group were all higher than 0.969 (Figure 2A).”

  1. Line 226 - Significantly differ

Response:

“Hormone receptor genes (FSHR, LHCGR, ESR1), steroid hormone secretion genes (CYP11A1, CYP17A1, CYP19A1, STAR, HSD3B, HSD17B1), and genes related to oocyte quality (PTX3, BCL2, GDF9, DNMT1, CASP3, IGFBP4) whose expression levels were significant differences between the ovaries of lambs and adult sheep (FDR<0.05) (Table 3).” was corrected as “In Table3, the expression levels of hormone receptor genes (Follicle Stimulating Hormone Receptor, FSHR; Luteinizing Hormone/Choriogonadotropin Receptor, LHCGR; Estrogen Receptor 1, ESR1), steroid hormone secretion genes (Cytochrome P450 family 11 subfamily A member 1, CYP11A1; cytochrome P450 family 17 subfamily A member 1, CYP17A1; cytochrome P450 family 19 subfamily A member 1, CYP19A1; steroidogenic acute regulatory protein, STAR; 3 beta-hydroxysteroid dehydrogenase, HSD3B; 17-beta-hydroxysteroid dehydrogenase type 1, HSD17B1), and genes related to oocyte quality (Pentraxin 3, PTX3; BCL2 apoptosis regulator, BCL2; growth differentiation factor 9, GDF9; DNA methyltransferase 1, DNMT1, Caspase 3, CASP3; insulin like growth factor binding protein 4, IGFBP4) were significantly different between the ovaries of lambs and adult sheep (FDR < 0.05).”

  1. Figure 2 - Not clear/visible

Response:

The Figure 2 has been enlarged.

  1. Line 245 and 247-250 - Needs grammatical correction

Response:

“In Supplementary file 7, there were 2194 genes were predicted targeted by the 358 differentially expressed miRNAs. The differentially expressed miRNAs which targeting key genes (FSHR, LHCGR, CYP11A1, STAR, PTX3, and ESR1) were listed in Table 4 and Figure 3. Oar-miR-143 which targeted FSHR with the high level and differential expression in ovaries between lamb and adult sheep [33, 34], and PTX3 gene which related to oocyte quality targeted by oar-miR-377-3p, oar-miR-485-3p, oar-miR-487a-3p, miR-496-x, and miR-767-x should be noticed. The most eye-catching is the miRNA–mRNA network related to the ESR1 and PTX3 gene (Figure 3).” was corrected as “As shown in Supplementary File 7, 2194 genes were predicted to be targeted by the 358 differentially expressed miRNAs. The differentially expressed miRNAs that targeted key genes (FSHR, LHCGR, CYP11A1, STAR, PTX3, and ESR1) are listed in Table 4 and Figure 3. Oar-miR-143, which targeted FSHR at a high level and with differential expression in ovaries between lamb and adult sheep [33, 34], and PTX3 gene, which was related to oocyte quality and targeted by oar-miR-377-3p, oar-miR-485-3p, oar-miR-487a-3p, miR-496-x, and miR-767-x, should be noticed. The most notable is the miRNA–mRNA network, related to ESR1 and PTX3 gene (Figure 3).”

  1. Line 262 - Difficult to read. It should be restructured as

…. Groups: a total of 21,194 target genes were mapped to GO in terms of the cellular component; 19,819 to molecular function and 21,888 to biological processes.

Response:

“there were 21,194, 19,819 and 21,888 target genes mapped to the GO terms of cellular component (CC), molecular function (MF) and biological processes (BP), respectively(Supplementary file 8)” was corrected as “a total of 21,194 target genes were mapped to GO in terms of the cellular component (CC): 19,819 to molecular function (MF) and 21,888 to biological processes (BP) (Supplementary File 8).”

  1. LINE 284 - Not readable. It should be “A total of 1,944, 1,763 and 2,004 differentially…….”

Response:

“There were 1,944, 1,763 and 2,004 differentially expressed mRNAs were separately annotated into the ontologies of biological process (BPs), cellular component (CCs) and molecular function (MFs).” was corrected as “A total of 1944, 1763 and 2004 differentially expressed mRNAs were respectively annotated in the GO terms of BP, CC, and MF.”

  1. Line 343 - “expression in lamb ovaries were highly significantly higher than”. These types of errors are all over the manuscript.

Response:

“However, the exciting finding was that FSHR and LHCGR expression in lamb ovaries were highly significantly higher than that in adult sheep ovaries.” was corrected as “However, the exciting finding was that FSHR and LHCGR expression in lamb ovaries was significantly higher than that in adult sheep ovaries.”

Thank you for showing our errors in the written English language point by point. We have tried our best to revise the manuscript according to the comments. We also send the manuscript for language revision by a professional editing company (MDPI English Editing Service).

Reviewer 2 Report

Female reproductive technologies such as juvenile in vitro fertilization and embryo transfer (JIVET) have been shown to accelerate genetic gain by increasing selection intensity and decreasing generation interval. However, oocytes obtained from juvenile donors show lower embryonic developmental potential when compared to adults’ gametes. The manuscript “Study on MiRNAome and Transcriptome in Ovine Ovary to Reveal the Mechanism of Poor Efficiency in Juvenile Superovulation’’ by Zhang and colleagues, analyzed the expression patterns of miRNA and mRNA in the ovaries of lamb and adult sheep underwent superovulation protocols using high-throughput sequencing technology. To do this, they used conventional superovulation protocols for sheep species in lambs and adult animals. Then, they extracted jugular-blood to measure serum levels of FSH, LH, P4, and E2, and the ovaries to analyze the production of follicles and their transcriptomic pattern (miRNA and mRNA), in order to elucidate molecular markers associated with the lower embryonic developmental potential of oocytes obtained from juvenile donors.

Thus, this study has relevance for molecular reproductive biology and animal production. The overall study is interesting, and the experiments well developed. However, the paper has many spelling errors, and the global grammar needs a deep correction. With the current phrasing is very difficult to perform a precise judgment of the present study. The English language should be improved to ensure that an international audience can clearly understand your text. In addition, some sentences are not in accordance with the cited reference (some of them are indicated below).

Both, the research question and how the research fills an identified knowledge gap are not well defined. However, it is important to mention that all underlying data have been provided, they are robust, statistically sound, & controlled. Also, the figures have good quality and well presented. Finally, some conclusions are not fully consistent with the evidence and arguments presented, as well it sounds like a repetition of the results.

Here there are the specific comments raised by this reviewer

Simple summary and abstract sections

FSHR et. al should not be used.

Introduction

1.- Introduction needs improvements. The research question and how the research fills an identified knowledge gap are not well defined.

Lines 45-46: Ambiguous sentence.

Line 51: please, check the accordance of the sentence with the referenced article (No. 4). This study investigated potential epigenetic mechanisms leading to the loss of oocyte development potential in young ewes by determining whether global genomic methylation changes were associated with donor age

Line 53: please, check/correct the sentence which is in past tense but then it changes to the present tense.

Line 56: It is difficult to understand. Please, correct the sentence or explain better what you are meaning with ''JIVET is not smooth''?

Line 54-57: I suggest including reference(s)

Lines 58-59: I suggest delete this sentence.

Lines 68-69: it is difficult to understand. Please, correct the sentence ‘’estrus trait formation’’?

Lines 70-71: it is difficult to understand. Please, improve grammar.

Lines 79-82: it is difficult to understand. Please, improve grammar.

Material and methods

Please provide references in each protocol/methodology

Since sheep reproduction reproductively seasonal animals, please, indicate the season where the study was performed.

Results

171-177: very difficult to understand

Line 180: This sentence can be omitted (I suggest delete it)

Discussion

Lines 314-316: Please, provide references

Lines 320-322: very difficult to understand

Lines 348-350: This sentence can be omitted (I suggest delete it)

Line 364-365: Please, check that the statement is in accordance with the reference. Reference 57 investigated the potential epigenetic mechanisms improving cumulus cells quality of prepubertal lambs by determining whether melatonin treatment altered gene expression of key enzymes associated with methylation modification. Thus, it is not in accordance with the sentence.

Line 367: the evidence showed here is not enough to conclude this.

Line 373. The reference has missed the format.

Lines 378-380: difficult to understand.

Author Response

Reviewer 2: 

This study has relevance for molecular reproductive biology and animal production. The overall study is interesting, and the experiments well developed. However, the paper has many spelling errors, and the global grammar needs a deep correction. With the current phrasing is very difficult to perform a precise judgment of the present study. The English language should be improved to ensure that an international audience can clearly understand your text. In addition, some sentences are not in accordance with the cited reference (some of them are indicated below).

Both, the research question and how the research fills an identified knowledge gap are not well defined. However, it is important to mention that all underlying data have been provided, they are robust, statistically sound, & controlled. Also, the figures have good quality and well presented. Finally, some conclusions are not fully consistent with the evidence and arguments presented, as well it sounds like a repetition of the results.

Comments:

  1. FSHR et. al should not be used

Response:

“FSHR et. al” was corrected asFSHR molecules, among others,”.

  1. Introduction needs improvements. The research question and how the research fills an identified knowledge gap are not well defined.

Response:

Combined with the comments from reviewers, we improved the introduction. And research question and how the research fills an identified knowledge gap are defined in the introduction: We supposed that the level of ovarian development in juvenile animals may be related to miRNA and mRNA expression, which contribute to the limitation of its application on a large scale.

  1. Lines 45-46: Ambiguous sentence.

Response:

“For breeding of domestic animals (such as sheep, cattle, pigs, etc.), limiting factors such as reproductive cycle and generation interval affect their genetic progress.” was corrected as “For breeding of domestic animals (sheep, cattle, pigs, etc.), limiting factors such as the reproductive cycle and generation interval affect their genetic progress.”.

  1. Line 51: please, check the accordance of the sentence with the referenced article (No. 4). This study investigated potential epigenetic mechanisms leading to the loss of oocyte development potential in young ewes by determining whether global genomic methylation changes were associated with donor age

Response:

Thank you for showing our misquotation. Here, we replace the citation with reference 1.

  1. Line 53: please, check/correct the sentence which is in past tense but then it changes to the present tense.

Response:

“Then the oocyte was matured and fertilized in vitro, and the produced embryos are transplanted into the recipient.” was corrected as “Then the oocytes are matured and fertilized in vitro, and the produced embryos are transplanted into the recipients.”.

  1. Line 56: It is difficult to understand. Please, correct the sentence or explain better what you are meaning with ''JIVET is not smooth''?

Response:

We add the content of “it was found that embryos derived from juvenile oocytes showed lower rates of development” to explain “JIVET is not smooth”.

  1. Line 54-57: I suggest including reference(s). Lines 58-59: I suggest delete this sentence.

Response:

First, we delete the sentence of “Recently, we speculated that the expression of microRNA (miRNA) and messenger RNA (mRNA) in juvenile ovaries may be responsible for the above issue.”.

Second, “Compared with the technique of Multiple Ovulation and Embryo Transfer (MOET) in adult females, the above process that so-called Juvenile in Vitro Embryo Transfer (JIVET) is not smooth, the level of ovarian development in juvenile animals may contribute to the limitation of JIVET application on a large scale.” was corrected as “Compared with the technique of multiple ovulation and embryo transfer (MOET) in adult females, the above process so-called juvenile in vitro embryo transfer (JIVET), is not smooth, and it was found that embryos derived from juvenile oocytes showed lower rates of development [5]. We supposed that the level of ovarian development in juvenile animals may be related to miRNA and mRNA expression, which contribute to the limitation of its application on a large scale.”

  1. Lines 68-69: it is difficult to understand. Please, correct the sentence ‘’estrus trait formation’’?

Response:

Another reviewer also raised the problem that the sentences here are too long and not easy to understand. Then, we have simplified this sentence. We changed the sentence as follows:

“In recent years, many studies have further shown that miRNAs are involved in mammalian follicular, oocyte and luteal development [15,16], ovarian function [17], proliferation and apoptosis of ovarian granulosa cells [18], female fertility and reproduction [19], pituitary gonadotropin (FSH and LH) secretion [20], estrogen action [21], prolificacy [22] and seasonal estrus trait formation [23,24].” was corrected as “In recent years, many studies have further shown that miRNAs are involved in mammalian follicular, oocyte and luteal development [15,16].”

  1. Lines 70-71: it is difficult to understand. Please, improve grammar.

Response:

 “Hereafter, the role of miRNAs in animal ovary has been constantly concerned by researchers.” was corrected as “Thereafter, the role of miRNAs in animal ovaries has been a constant concern of researchers.”

  1. Lines 79-82: it is difficult to understand. Please, improve grammar.

Response:

 “Recently, ovine gene expression in prepubertal and adult super-stimulated follicle granulosa cells were studied using RNA sequencing technology, more than 300 differentially expressed genes were reported, and beta-estradiol upstream regulator were concerned in prepubertal ovary [32].” was corrected as “Recently, ovine gene expression in prepubertal and adult superstimulated follicle granulosa cells were studied using RNA sequencing technology, and more than 300 differentially expressed genes were reported; beta-estradiol upstream regulator was a concern in prepubertal ovary [32]”

  1. Please provide references in each protocol/methodology

Response:

 Some references were cited for protocol/methodology in Materials and Methods.

  1. Since sheep reproduction reproductively seasonal animals, please, indicate the season where the study was performed.

Response:

The experiment in this study was carried out in the spring. We added this information in the part of 2.1.

  1. 171-177: very difficult to understand

Response:

This paragraph was corrected as “Under the stimulation of exogenous hormone, superovulation was successfully performed in lambs and adult Hu sheep. The follicle numbers and serum reproduction hormones are presented in Table 1. The mean superstimulated follicle number (total of two ovaries) in lambs was 70.17 ± 5.14, which was highly significantly (p < 0.01) and much more than that in adult sheep (20.17 ± 2.65). However, the concentrations of FSH, LH, progesterone (P4), and estradiol (E2) in jugular vein serum all had no significant difference between lamb and adult groups after superovulation treatment.”

  1. Line 180: This sentence can be omitted (I suggest delete it)

Response:

Thank you for your suggestion.

  1. Lines 314-316: Please, provide references

Response:

We have provided the references after the statement.

  1. Lines 320-322: very difficult to understand

Response:

This sentence was corrected as “In the present study, we focused on the initial stage of JIVET, ovaries under the treatment of superovulation, therefore miRNAome and transcriptome were compared between juvenile and adult ovaries to discover the mechanism of inferior oocyte capture by JIVET.”

  1. Lines 348-350: This sentence can be omitted (I suggest delete it)

Response:

We have deleted the sentence.

  1. Line 364-365: Please, check that the statement is in accordance with the reference. Reference 57 investigated the potential epigenetic mechanisms improving cumulus cells quality of prepubertal lambs by determining whether melatonin treatment altered gene expression of key enzymes associated with methylation modification. Thus, it is not in accordance with the sentence.

Response:

Thank you for showing our misquotation. Here, we replace the citation with reference of “Palma, G.A.; Tortonese, D.J.; Sinowatz, F., Developmental capacity in vitro of prepubertal oocytes. Anat. Histol. Embryol. 2001, 30, 295-300.”.  

  1. Line 367: the evidence showed here is not enough to conclude this.

Response:

“Therefore, the down-regulated PTX3 mRNA in ovaries of lamb suggested that the lamb could not be a good quality oocyte donor.” was corrected as “For the present study, the down-regulated PTX3 mRNA in ovaries of lamb suggested that oocytes derived from lamb may be of poor quality.”

  1. Line 373. The reference has missed the format.

Response:

Under the guidance of a reviewer, we deleted this sentence.

  1. Lines 378-380: difficult to understand.

Response:

This sentence was corrected as “DNMT1 is responsible for maintaining methylation patterns throughout DNA replication, and we found its mRNA expression is higher in lamb ovaries. However, Fang et al. had found that CoCs derived from lamb matured in vitro with a lower mRNA expression of DNMT1 [23]. This suggests that CoCs and ovarian tissue of lamb may be inconsistent in methylation levels. CASP3 involved in apoptosis had been proved responsible for follicular atresia [62], a higher level mRNA expression of CASP3 in lamb ovaries for this study may reflect that a more follicular atresia event is happening. IGFBP4 mRNA transcripts were detected in human follicular fluid, and had been demonstrated that was associated with oocyte maturation and embryo development [63]. This suggests that lower mRNA expression level of IGFBP4 detected in lamb ovaries may lead to the emergence of low-quality oocytes.”

Reviewer 3 Report

General comments: The present manuscript covers two very interesting topics: the poor superovulatory response in prepubertal sheep and the integration of miRNA evaluation into the transcriptome analysis of assisted reproductive technologies like JIVET. This is especially interesting in small ruminants such as sheep. As authors have described, RNA sequencing technology have already been applied to the study of transcript abundance in ovine granulosa cells obtained from ovaries of prepubertal and adult sheep after hormonal stimulation (FSH), but that study only covered mRNAs, so, in my opinion, integrated analysis of miRNA and mRNA should be of high interest. If, in fact, it is the first time this kind of analysis is performed in ovaries from prepubertal ewes, it should be highlighted throughout the text to a greater extent.

The results obtained in regard to FSHR and LHCGR mRNA being highly expressed in lamb ovaries together with the fact that ESR1 (an estrogen receptor) was down-regulated is highly interesting. Furthermore, authors managed to identify a series of miRNA that could be exerting their function on ESR1, suppressing it. That would help increase our understanding of how estradiol/ESR1 affects the ovary of prepubertal and adult sheep in a different way.

Nevertheless, the main issue I had with this manuscript was reading it itself. Some sentences were not clear or were quite hard to understand because of the language used. Therefore, to be considered for publication, extensive editing of English language will be necessary.

Moreover, information about sample collection is missing. It is not clear if authors have used just granulosa cells or not. Also, I have an issue regarding the terminology of ‘gene expression’ in the text. Genes are said to be expressed when the proteins they encode function in an organism. This means that, in addition to transcription, ‘gene expression’ includes protein translation. Therefore, ‘gene expression’ should be defined as the whole process, from gene to the functional gene product. I know it is a common mistake and everyday one gets to see papers which claim to analyze gene expression when performing microarrays, qPCR or RNA seq. In my humble opinion, these methods determine mRNA levels, that is, the template of a protein, nothing else. Hence, care must be taken when using these terms.

Simple summary and abstract:

Lines 22 and 36: Authors only mention oar-miR-485-3p and oar-miR-377-3p. What about oar-miR-487a-3p?

Introduction:

Line 77: I think this sentence would have more sense in the first paragraph.

Line 78: This sentence is not clear. Do authors mean mRNA maternal storage in the oocyte? In any case, please delete it since it does not seem relevant for this study.

Lines 79-84: These two sentences should be rewritten and joined together. First, both studies used RNA seq technology to evaluate the hormone response of prepubertal and adult sheep ovaries. The main difference is that authors have now explored miRNAs as well.

Line 86: Authors say that their study will “reveal the information mechanism of large number of follicles and poor oocyte quality…” but have not actually studied oocyte quality. This  study  did  not  focus  on  oocyte  quality and developmental competence, so no conclusions can be made in that  direction. Please correct or clarify the meaning.

Materials and methods:

Line 93: Pregnant mare serum gonadotropin (PMSG).

Line 105: Change “RCF” for “x g”.

Line 106:  Progesterone (P4) and estradiol (E2). Delete from line 175 in the results section. Also, there is information missing about the determination of FSH, LH, P4 and E2 concentrations and the ovariectomy procedure.

Line 107: Sampling and RNA extraction protocol is completely missing and I should be explained in detail because it is the basis of the experimental design.

Line 110: What was the method to fragment the RNA?

Lines 111-114: Information about cDNA synthesis and PCR is missing. Please correct.

Line 139: What was the rationale behind using that many software packages?

Line 155: Please define all gene names and write symbols in italics, e.g. Estrogen Receptor 1 (ESR1).

Line 167: For proper understanding, a section dedicated only to the statistical analysis should be included. Please correct it and add the statistics of all the evaluations performed.

Results:

Line 178: Table 1: Please define what the “Number” column refers to. In the third column, specify it is the average of the number of follicles.

Line 210: Figure 1: The format of this figure should be enhanced for proper reading since some names are too small to see. Also, please add x- and y-axis titles that are missing.

Line 214: Table 2: What does the “-x” mean in some miRNA?

Line 223: Out of the 3150 differentially expressed mRNAs between lamb and adult ovaries, the reason why authors chose to show the ones in Table 3 should be noted.

Line 234: Table 3: An extra column defining the groups in which those genes are divided (as stated in lines 223-225) would be good for ease of reading.

Line 234: Table 4: Same as in table 3, an extra column would be appreciated.

Line 245: Did those 2194 mRNAs come from the 3150 differentially expressed ones or the total ones (21,352)?

Line 246: What was the rationale behind the use of those key genes and not others?

Line 311: Figure 4: The y-axis title is missing (Relative abundance of mRNA transcripts).

Discussion:

Line 317: Authors claim that the poor efficiency of JIVET has been attributed to “maturation status of cumulus cell [39], oocyte cytoplasm [40], oocytes nuclear [41], and follicular fluid microenvironment in juvenile ovary [42-44]”. Yet, RNA seems to have been extracted from whole ovaries in the present study. If that is the case, would not it be more convenient to evaluate specific cell types such as cumulus cells, mural, oocytes, etc?

Line 325: In order to facilitate reading, I would advise that, either here or in the introduction, it is mentioned that miRNAs negatively regulate gene expression at a post-transcriptional level, in case non-expert readers might get lost when they read the word “suppressed”.

Line 329: This sentence is not clear. I would assume that the final goal is to associate a higher abundance of miRNA transcripts in prepubertal ovaries (present study) with a lowe abundance of mRNAs and proteins in the same ovaries (Wu et al.), but this should be done with caution since no protein levels have been investigated in this study. Moreover, why did not include the mRNA results as well as miRNA? In this regard, both studies showed a greater number of down-regulated genes in terms of decreased transcription and so that could be a similarity.

Line 334: Authors suggest that “the lamb stage of ovaries might not be suitable for the synthesis of progesterone and estrogen”, but blood levels of progesterone and estradiol did not show significant differences between lambs and adult ewes. How can this be explained? Furthermore, it is later said (line 355) that “…the mRNAs expression level for steroid hormone or estradiol synthesis (CYP11A1, CYP17A1, CYP19A1, STAR, HSD3B, HSD17B1) in ovaries of lamb were as well significantly higher than that in adult sheep”. In my opinion, the issue here is more like, estradiol synthesis is happening, but there is an inhibition at the level of its receptors (ESR1). This suppression could be happening through the up-regulation of miRNA transcripts.

Line 335: For ease of reading, remind that those results (down-regulation of those miRNA transcripts) were obtained in the current study.

Lines 357-363: Taken into consideration the importance of ESR1 in the results obtained in this study, additional information may be provided. For example, a novel role in regulating oocyte meiotic resumption using follicular granulosa cells has been recently described (10.1038/cddis.2017.82). Moreover, “in human and mouse cumulus cells downregulation of ESR1 could be related to oocyte competence and is likely to be the driver of expression changes highlighted in the PI3K/AKT pathway” (https://doi.org/10.1093/humrep/dex320).

Lines 369-372: I disagree with these statements. In sheep oocytes, it has been previously demonstrated that GDF9 mRNA transcripts are quite detectable after IVM (https://doi.org/10.1016/j.theriogenology.2011.07.007; https://doi.org/10.3390/ani10050763). They might decrease compared to earlier stages ( 10.1016/j.theriogenology.2015.09.022) but are still transcribed. Authors are basing their GDF9 mRNA results (greater levels in lamb vs ewe) on a study in canine oocytes, a species which displays very uncommon features of meiosis resumption. In fact, “In this species, the expression of GDF9 and BMP15 proteins has been reported to decline in follicular cells with increasing follicle size during estrus, possibly in a species-specific manner related to the delay in cumulus expansion after ovulation” (10.3390/cells9041002). Also, please fix the citations here since 59 and 60 are basically referring to the same paper.

Moreover, it has been demonstrated that “GDF9 maintains the integrity of the cumulus-oocyte complex in preovulatory follicles and after ovulation, particularly by promoting the synthesis of PTX3” (10.3390/cells9041002). Since, in the present study, PTX3 mRNA relative abundance in prepubertal lambs was lower than that in adult ewes, but GDF9 mRNA was actually higher, could it be that the mechanism through which oocyte quality may be compromised was independent of GDF9? This might be associated with the miRNA-target mRNA interactions the authors have studied.

Line 373: I do not see the point of comparing immature and mature cumulus cells when the important issue here is young vs adult animals.

Line 375: Please determine what kind of differential expression that is. In this case, there was a greater abundance of BCL2 mRNA transcript in adult compared to prepubertal ovaries.

Line 378: I agree that the ratio of BAX/BCL2 determines the susceptibility of cells to apoptosis but, was this ratio really studied? In order to be drawn towards apoptosis rather than survival, BAX needs to be higher than BCL2.

Line 379: It would be appreciated if authors briefly explain the main functions of those genes (DNMT1, CASP3 and IFGBP4) in the ovary. Caspase-3 is expressed in many cell types, not just the oocyte (since it is the only cell described here) and, in bovine, IGFBP4 mRNA transcripts were detected only in cumulus cells, not the oocyte (https://doi.org/10.1016/j.domaniend.2004.03.003).

Line 384: This is very interesting.

Line 387: I would highlight the fact that those miRNAs are species-specific (sheep).

Line 390: A cite (the study by Zhang et al.) is missing after the word “production”.

Line 394: Very interesting.

Lines 397-411: I think this paragraph will gain a lot after the English language is revised.

Conclusions: The same goes for this section. Language editing will help improve the understanding of these conclusions. Furthermore, why did authors not include oar-miR-487-3p here? The same happened at the beginning in the simple summary and abstract.

Final remarks: As previously stated, this study is really interesting and I am generally positive about this manuscript. Nonetheless, there is room for improvement, especially in the Discussion section and it must be revised.

Author Response

Reviewer 3: 

The present manuscript covers two very interesting topics: the poor superovulatory response in prepubertal sheep and the integration of miRNA evaluation into the transcriptome analysis of assisted reproductive technologies like JIVET. This is especially interesting in small ruminants such as sheep. As authors have described, RNA sequencing technology have already been applied to the study of transcript abundance in ovine granulosa cells obtained from ovaries of prepubertal and adult sheep after hormonal stimulation (FSH), but that study only covered mRNAs, so, in my opinion, integrated analysis of miRNA and mRNA should be of high interest. If, in fact, it is the first time this kind of analysis is performed in ovaries from prepubertal ewes, it should be highlighted throughout the text to a greater extent.

The results obtained in regard to FSHR and LHCGR mRNA being highly expressed in lamb ovaries together with the fact that ESR1 (an estrogen receptor) was down-regulated is highly interesting. Furthermore, authors managed to identify a series of miRNA that could be exerting their function on ESR1, suppressing it. That would help increase our understanding of how estradiol/ESR1 affects the ovary of prepubertal and adult sheep in a different way.

Nevertheless, the main issue I had with this manuscript was reading it itself. Some sentences were not clear or were quite hard to understand because of the language used. Therefore, to be considered for publication, extensive editing of English language will be necessary.

Moreover, information about sample collection is missing. It is not clear if authors have used just granulosa cells or not. Also, I have an issue regarding the terminology of ‘gene expression’ in the text. Genes are said to be expressed when the proteins they encode function in an organism. This means that, in addition to transcription, ‘gene expression’ includes protein translation. Therefore, ‘gene expression’ should be defined as the whole process, from gene to the functional gene product. I know it is a common mistake and everyday one gets to see papers which claim to analyze gene expression when performing microarrays, qPCR or RNA seq. In my humble opinion, these methods determine mRNA levels, that is, the template of a protein, nothing else. Hence, care must be taken when using these terms.

Response:

We added the details about sample collection in the materials and methods. And we have checked all of the statement about “gene expression” in the manuscript, some of words of “gene” was corrected as “mRNA”. Thank you for your advice.

Comments:

  1. Lines 22 and 36: Authors only mention oar-miR-485-3p and oar-miR-377-3p. What about oar-miR-487a-3p?

Response:

In the Real-time PCR validation for miRNA, we did not successfully verify the difference expression of oar-miR-487a-3p in two groups. Therefore, we removed the gene from the Simple Summary and Abstract.

  1. Line 77: I think this sentence would have more sense in the first paragraph.

Response:

Thank you for your advice, we moved the sentence to the first paragraph.

  1. Line 78: This sentence is not clear. Do authors mean mRNA maternal storage in the oocyte? In any case, please delete it since it does not seem relevant for this study.

Response:

Thank you for your advice, we delete the sentence.

  1. Lines 79-84: These two sentences should be rewritten and joined together. First, both studies used RNA seq technology to evaluate the hormone response of prepubertal and adult sheep ovaries. The main difference is that authors have now explored miRNAs as well.

Response:

(a) Follicle granulosa cells were selected as the research materials in Wu’s study, while the whole ovaries (including granulosa cells, oocyte, theca cells and so on) were studied in our research.

(b) Wu et al. had not presented their research data of serum hormone and follicles numbers in super-stimulated juvenile and adult sheep.

(c) For ease of reading, we combine the two sentences of “In the present study……” and “Differentially expressed……” into one sentence.

  1. Line 86: Authors say that their study will “reveal the information mechanism of large number of follicles and poor oocyte quality…” but have not actually studied oocyte quality. This study did not focus on oocyte quality and developmental competence, so no conclusions can be made in that direction. Please correct or clarify the meaning.

Response:

Many studies have shown that oocytes derived from juvenile animals have poor subsequent development [22]. In 2015, our team also reported that the low global DNA methylation and hydroxymethylation in oocytes derived from lambs associated with subsequent lesser developmental potential [23]. Therefore, the poor developmental potential of juvenile oocytes is a proven fact, and subsequent developmental abilities were not reassessed in this study.

We add the above content to the beginning of the third paragraph as a necessary explanation.

  1. Line 93: Pregnant mare serum gonadotropin (PMSG).

Response:

Thank you for point out our omission of abbreviations. And we've made the change.

  1. Line 105: Change “RCF” for “x g”.

Response:

“1500 RCF” was corrected as “1500 × g”.

  1. Line 106: Progesterone (P4) and estradiol (E2). Delete from line 175 in the results section. Also, there is information missing about the determination of FSH, LH, P4 and E2 concentrations and the ovariectomy procedure. Line 107: Sampling and RNA extraction protocol is completely missing and I should be explained in detail because it is the basis of the experimental design.

Response:

(a) “the product of serum was used for FSH, LH, P4 and E2 concentration determination” was corrected as “the product of serum was used for FSH, LH, progesterone (P4) and estradiol (E2) concentration determination”.

(b) We hope to show that there is no significant difference in reproductive hormone levels between the two groups, but there were significant differences in the mRNA expression of hormone receptor. Therefore, we believe that the ability to receive reproductive hormone signals differs between the two groups. The above discussion content had been presented in the third paragraph of the Discussion.

(c) The serum hormones of FSH, LH, E2 and P4 were measured according to the instructions of Iodine [125I] radioimmunoassay kit (BNIBT, Beijing, China). The above content is added in the part of 2.2.

(d) Three sheep were randomly selected from each group for anesthesia, then the abdominal skin and muscle were cut open with 5cm length by a scalpel, and the left ovary was pulled out of the body. After tubal ligation, the left ovary was removed and frozen in liquid nitrogen instantly. In the liquid nitrogen freezing environment, each whole ovary was ground into powder. Finally, the powder was collected for RNA extraction using Trizol method. The above content is added in the part of 2.2.

  1. Line 110: What was the method to fragment the RNA? Lines 111-114: Information about cDNA synthesis and PCR is missing. Please correct.

Response:

The method to fragment the RNA and information about cDNA synthesis were added in the part of 2.3 as follows:

Small RNA in a size range of 18-30 nt were isolated from total RNA based on 15% denaturing polyacrylamide gel electrophoresis (PAGE). Then the 3’ adapters were added and the 36-44nt RNAs were enriched. The 5’ adapters were then ligated to the RNAs as well. The ligation products were reverse transcribed by PCR amplification and the 140-160bp size PCR products were enriched to generate a cDNA library.

  1. Line 139: What was the rationale behind using that many software packages?

Response:

The rationale of these software packages: The 2-8nt sequences which start from 5’ end of miRNA were choose as seed sequences to predict with 3’-UTR of target mRNAs.

Then, “RNAhybrid (Version 2.1.2), svm_light (Version 6.01), Miranda (Version 3.3a) and TargetScan (Version 7.0) were used for miRNA targets predicting” was corrected as “The 2-8nt sequences which start from 5’ end of miRNA were choose as seed sequences to predict with 3’-UTR of target mRNAs. RNAhybrid (Version 2.1.2), svm_light (Version 6.01), Miranda (Version 3.3a) and TargetScan (Version 7.0) were used for miRNA targets predicting. The intersection of the above results were more credible to be chosen as predicted miRNA target genes”.

  1. Line 155: Please define all gene names and write symbols in italics, e.g. Estrogen Receptor 1 (ESR1).

Response:

Thank you. We define all gene names and write symbols in italics.

  1. Line 167: For proper understanding, a section dedicated only to the statistical analysis should be included. Please correct it and add the statistics of all the evaluations performed.

Response:

We add a part of “2.8 Statistical analysis”:

The chi-square test was used to analyze categorical variables of follicles number. Duncan’s multiple range test program in ANOVA was adopted to analyze continuous variables of hormone concentration and mRNA/miRNA expression based on the SAS 8.0 software (SAS Institute Inc., North Carolina, USA). All of the results were presented with mean ± SE. To assess the relationship between high-throughput sequencing and real-time PCR in mRNA/miRNA expression assay, the Spearman correlation was calculated based on SPSS version 20.0 (SPSS, Inc., IL, USA)).

  1. Line 178: Table 1: Please define what the “Number” column refers to. In the third column, specify it is the average of the number of follicles.

Response:

In Table 1, “Number” was corrected as “Statistics number”, and “Follicles number” was corrected as “Average number of follicles”. 

  1. Line 210: Figure 1: The format of this figure should be enhanced for proper reading since some names are too small to see. Also, please add x- and y-axis titles that are missing.

Response:

Thank you for your advice, Figure 1 has been enlarged, and the titles of the X and Y-axis in Figure 1 have also been added.

  1. Line 214: Table 2: What does the “-x” mean in some miRNA?

Response:

Prefix of oar-miR represents that the miRNA has been included in miRBase for species of ovis aries, while without of the prefix represents that the miRNA has not been included in the data of ovis aries. Moreover, “miR-number-x” and “miR-number-y” represent that the sequence can be aligned to the -5p and -3p for the corresponding miRNA in miRBase for other animal species.

We add the above content to the materials and methods (2.4).

  1. Line 223: Out of the 3150 differentially expressed mRNAs between lamb and adult ovaries, the reason why authors chose to show the ones in Table 3 should be noted.

Response:

In order to pay attention to the differences in the sensitivity of juvenile and adult sheep to superovulation hormones, and key genes expression related to oocyte quality, we selected some genes which presented in Table 3 to emphasize their expression differences between the two groups.

We add the above content to the results part (3.3).

  1. Line 234: Table 3: An extra column defining the groups in which those genes are divided (as stated in lines 223-225) would be good for ease of reading.

Response:

We added an extra column to define the groups in which those genes are divided in Table 3.

  1. Line 234: Table 4: Same as in table 3, an extra column would be appreciated.

Response:

We also added an extra column to define the groups in which those genes are divided in Table 4.

  1. Line 245: Did those 2194 mRNAs come from the 3150 differentially expressed ones or the total ones (21,352)?

Response:

These 2194 mRNAs come from the 3150 differentially expressed mRNAs. The details can be seen in Supplementary file 7.

  1. Line 246: What was the rationale behind the use of those key genes and not others?

Response:

Considering that these genes have been shown to be related to hormone response or oocyte quality for many times in previous studies, we believe that taking these genes out for separate analysis is conducive to the study of corresponding miRNA.

  1. Line 311: Figure 4: The y-axis title is missing (Relative abundance of mRNA transcripts).

Response:

The y-axis title for Figure 4 was added.

  1. Line 317: Authors claim that the poor efficiency of JIVET has been attributed to “maturation status of cumulus cell [39], oocyte cytoplasm [40], oocytes nuclear [41], and follicular fluid microenvironment in juvenile ovary [42-44]”. Yet, RNA seems to have been extracted from whole ovaries in the present study. If that is the case, would not it be more convenient to evaluate specific cell types such as cumulus cells, mural, oocytes, etc?

Response:

A number of factors may contribute to JIVET's inefficiency, including cumulus cells, oocytes, and oocyte nuclear maturation status. Yes, studying a specific cell in the ovary may help explain its role in influencing JIVET’s inefficiency. Thank you for your advice, we will use single-cell sequencing technology to study this topic in the near future.  

  1. Line 325: In order to facilitate reading, I would advise that, either here or in the introduction, it is mentioned that miRNAs negatively regulate gene expression at a post-transcriptional level, in case non-expert readers might get lost when they read the word “suppressed”.

Response:

We add the content of “Considering that miRNAs can negatively regulate genes expression at the post-transcriptional level, we speculate that a much more genes expression in lamb ovaries were suppressed.” in this paragraph.

  1. Line 329: This sentence is not clear. I would assume that the final goal is to associate a higher abundance of miRNA transcripts in prepubertal ovaries (present study) with a lowe abundance of mRNAs and proteins in the same ovaries (Wu et al.), but this should be done with caution since no protein levels have been investigated in this study. Moreover, why did not include the mRNA results as well as miRNA? In this regard, both studies showed a greater number of down-regulated genes in terms of decreased transcription and so that could be a similarity.

Response:

“The results from researches of Wu et al. bears out our findings in miRNAs comparing.” was corrected as “In our results of mRNAs, we found 1791 downregulated mRNAs vs. 1359 upregulated mRNAs in ovaries of lamb as well. The results from researches of Wu et al. and our mRNA results verify the findings in comparing miRNAs.”

  1. Line 334: Authors suggest that “the lamb stage of ovaries might not be suitable for the synthesis of progesterone and estrogen”, but blood levels of progesterone and estradiol did not show significant differences between lambs and adult ewes. How can this be explained? Furthermore, it is later said (line 355) that “…the mRNAs expression level for steroid hormone or estradiol synthesis (CYP11A1, CYP17A1, CYP19A1, STAR, HSD3B, HSD17B1) in ovaries of lamb were as well significantly higher than that in adult sheep”. In my opinion, the issue here is more like, estradiol synthesis is happening, but there is an inhibition at the level of its receptors (ESR1). This suppression could be happening through the up-regulation of miRNA transcripts.

Response:

(a) “These results suggest that ovaries in the lamb stage might be not suitable for the synthesis of progesterone and estrogen.” was deleted.

(b) Then “In this study, the mRNAs expression level for steroid hormone or estradiol synthesis (CYP11A1, CYP17A1, CYP19A1, STAR, HSD3B, HSD17B1) in ovaries of lamb were as well significantly higher than that in adult sheep.” was corrected as “In this study, the mRNA expression levels of steroid hormone or estradiol synthesis (CYP11A1, CYP17A1, CYP19A1, STAR, HSD3B, HSD17B1) in lamb ovaries were also significantly higher than in adult sheep, which means that estradiol synthesis is happening.”

  1. Line 335: For ease of reading, remind that those results (down-regulation of those miRNA transcripts) were obtained in the current study.

Response:

“down-regulated miRNA of oar-miR-10a and miR-145-x in lamb were all speculated implicating in suppression of granulosa cell proliferation by targeting on brain-derived neurotropic factor (BDNF) and activin receptor IB (ACVRIB) [7,46,47].” was corrected as “downregulated miRNAs of oar-miR-10a and miR-145-x in lamb for the present study was speculated, implicating suppression of granulosa cell proliferation by targeting of brain-derived neurotropic factor (BDNF) and activin receptor IB (ACVRIB) [7,46,47].”

  1. Lines 357-363: Taken into consideration the importance of ESR1 in the results obtained in this study, additional information may be provided. For example, a novel role in regulating oocyte meiotic resumption using follicular granulosa cells has been recently described (10.1038/cddis.2017.82). Moreover, “in human and mouse cumulus cells downregulation of ESR1 could be related to oocyte competence and is likely to be the driver of expression changes highlighted in the PI3K/AKT pathway” (https://doi.org/10.1093/humrep/dex320).

Response:

Thank you very much for helping us to improve our discussion part.

The content of “Liu et al. showed that the estradiol (E2)-estrogen receptors (ERs) system in follicular granulosa cells has a dominant role in controlling oocyte meiotic resumption in mammals [56]. In human and mouse cumulus cells, downregulation of ESR1 could be related to oocyte competence and is likely to be the driver of expression changes highlighted in the PI3K/AKT pathway[57]”

  1. Lines 369-372: I disagree with these statements. In sheep oocytes, it has been previously demonstrated that GDF9 mRNA transcripts are quite detectable after IVM (https://doi.org/10.1016/j.theriogenology.2011.07.007; https://doi.org/10.3390/ani10050763) . They might decrease compared to earlier stages ( 10.1016/j.theriogenology.2015.09.022) but are still transcribed. Authors are basing their GDF9 mRNA results (greater levels in lamb vs ewe) on a study in canine oocytes, a species which displays very uncommon features of meiosis resumption. In fact, “In this species, the expression of GDF9 and BMP15 proteins has been reported to decline in follicular cells with increasing follicle size during estrus, possibly in a species-specific manner related to the delay in cumulus expansion after ovulation” (10.3390/cells9041002). Also, please fix the citations here since 59 and 60 are basically referring to the same paper. Moreover, it has been demonstrated that “GDF9 maintains the integrity of the cumulus-oocyte complex in preovulatory follicles and after ovulation, particularly by promoting the synthesis of PTX3” (10.3390/cells9041002). Since, in the present study, PTX3 mRNA relative abundance in prepubertal lambs was lower than that in adult ewes, but GDF9 mRNA was actually higher, could it be that the mechanism through which oocyte quality may be compromised was independent of GDF9? This might be associated with the miRNA-target mRNA interactions the authors have studied.

Response:

Combined with the references you mentioned, we think upregulation GDF9 in the lamb group may indicated that ovaries from lamb contain more primordial follicles than adult sheep. We made the following changes:

“GDF9 is supposed to promote progression of primary follicles to later preantral stages in mammal, and GDF9 was reported at high levels only in immature GV oocytes, but was barely detectable in metaphase II oocytes after in vitro maturation [59,60]. GDF9 up-regulated in lamb group indicated that the development level of superstimulated follicles in the lambs was lower than adults.” was corrected as “Kona et al. reported that the expression of GDF9 was highest in ovine primordial follicles compared with preantral, early antral, antral, and large antral stage [60]. Upregulation GDF9 in the lamb group may indicated that ovaries from lamb contain more primordial follicles than adult sheep.”

  1. Line 373: I do not see the point of comparing immature and mature cumulus cells when the important issue here is young vs adult animals.

Response:

The sentence of “Filali et al. found that BCL2 mRNA expression was significantly higher in cumulus cells for mature oocytes than for immature oocytes, and there was no significant difference in BAX expression in mature and immature cumulus cells [61].” was deleted.

  1. Line 375: Please determine what kind of differential expression that is. In this case, there was a greater abundance of BCL2 mRNA transcript in adult compared to prepubertal ovaries.

Response:

 “In this study, we also only found differential expression of BCL2 gene in the two groups but did not find the difference in BAX.” was corrected as “However, we found mRNA expression of BCL2 gene in lamb ovaries were downregulated, but did not find a difference in BAX.”

  1. Line 378: I agree that the ratio of BAX/BCL2 determines the susceptibility of cells to apoptosis but, was this ratio really studied? In order to be drawn towards apoptosis rather than survival, BAX needs to be higher than BCL2.

Response:

The information of BAX mRNA expression and the ratio of BAX/BCL2 are presented in Table 3 now. In this study, the mRNA expression of BAX is also higher than BCL2.

  1. Line 379: It would be appreciated if authors briefly explain the main functions of those genes (DNMT1, CASP3 and IFGBP4) in the ovary. Caspase-3 is expressed in many cell types, not just the oocyte (since it is the only cell described here) and, in bovine, IGFBP4 mRNA transcripts were detected only in cumulus cells, not the oocyte (https://doi.org/10.1016/j.domaniend.2004.03.003).

Response:

We briefly explained the main functions of DNMT1, CASP3 and IFGBP4 in ovary, and discussed according our results. Thank you for your advice, the follow content is presented in discussion part:

DNMT1 is responsible for maintaining methylation patterns throughout DNA replication, and we found its mRNA expression is higher in lamb ovaries. However, Fang et al. had found that CoCs derived from lamb matured in vitro with a lower mRNA expression of DNMT1 [23]. This suggests that CoCs and ovarian tissue of lamb may be inconsistent in methylation levels. CASP3 involved in apoptosis had been proved responsible for follicular atresia [62], a higher level mRNA expression of CASP3 in lamb ovaries for this study may reflect that a more follicular atresia event is happening. IGFBP4 mRNA transcripts were detected in human follicular fluid, and had been demonstrated that was associated with oocyte maturation and embryo development [63]. This suggests that lower mRNA expression level of IGFBP4 detected in lamb ovaries may lead to the emergence of low-quality oocytes.

  1. Line 384: This is very interesting. Line 387: I would highlight the fact that those miRNAs are species-specific (sheep).

Response:

Thank you for your advice.

“MiRNA of oar-miR-485-3p, oar-miR-487a-3p and oar-miR-377-3p which targets the PTX3 in ovine ovary, should be further studied in the future.” was corrected as “miRNAs of oar-miR-485-3p, oar-miR-487a-3p, and oar-miR-377-3p are species-specific of sheep, which target PTX3 in ovine ovary, should be further studied in the future.”

  1. Line 390: A cite (the study by Zhang et al.) is missing after the word “production”. Line 394: Very interesting.

Response:

The citation for the study by Zhang et al. is added after the word “production”.

  1. Lines 397-411: I think this paragraph will gain a lot after the English language is revised.

Response:

This paragraph have been revised by English native speaker from a professional editing company (MDPI English Editing Service).

  1. Conclusions: The same goes for this section. Language editing will help improve the understanding of these conclusions. Furthermore, why did authors not include oar-miR-487-3p here? The same happened at the beginning in the simple summary and abstract.

Response:

The manuscript in full was all revised by an English native speaker.

As previously stated, in the Real-time PCR validation for miRNA, we did not successfully verify the difference expression of oar-miR-487a-3p in two groups. Therefore, we removed the gene from the Simple Summary, Abstract, and the conclusion.

  1. Final remarks: As previously stated, this study is really interesting and I am generally positive about this manuscript. Nonetheless, there is room for improvement, especially in the Discussion section and it must be revised.

Response:

We appreciate your constructive comments and suggestions on our manuscript entitled. We have tried our best to revise our manuscript according to the comments. We would like to express our great appreciation to you for comments on our paper. Thank you.

Round 2

Reviewer 1 Report

Authors have addressed my comments and incorporated my suggestions. The manuscript is significantly improved. I think it can be published in present after reviewing for English languages/phrasal errors. 

Author Response

Reviewer 1: 

Authors have addressed my comments and incorporated my suggestions. The manuscript is significantly improved. I think it can be published in present after reviewing for English languages/phrasal errors.

Response:

Thank you for your advice and your previous comments.

Since English is not our native language, we send the manuscript again to the MDPI English Editing Company for a specialist edit. I hope this manuscript can be better presented after the specialist edit service.

Reviewer 2 Report

Although the authors have addressed most of the concerns previously raised by the reviewers, the manuscript is still requiring heavy editing to improve its English Language, not just for accuracy, but to allow the reader to be certain what is meant in numerous sections of the manuscript. I would strongly suggest language editing services. In conclusion, this reviewer can not consider the manuscript for publication in its current version.

Author Response

Reviewer 2: 

Although the authors have addressed most of the concerns previously raised by the reviewers, the manuscript is still requiring heavy editing to improve its English Language, not just for accuracy, but to allow the reader to be certain what is meant in numerous sections of the manuscript. I would strongly suggest language editing services. In conclusion, this reviewer can not consider the manuscript for publication in its current version.

Response:

Thank you for your suggestion and your previous comments. In the last revision of our manuscript, we had sent the manuscript to the MDPI English Editing Company for a regular edit, and now it's not effective enough. In this revision, we send the manuscript again to the MDPI English Editing Company for a specialist edit which we hope it can be better presented to readers after the specialist edit service.

Reviewer 3 Report

The authors have extensively revised and, for the most part, optimized the manuscript according to my comments. Therefore, I find the manuscript publishable. Nonetheless, the quality of all figures (which was asked to be enhanced) is worse than in the first manuscript and should be corrected prior to publication.

Also, just a few comments:

  • Line 167: Please define gene names.
  • Line 194, Table 1: Please delete "Statistics number". Something like "Number of animals", "Number of subjects", etc. would be more appropriate.
  • Figures 1, 2 and 3: It is impossible to read whats in the figures, please fix them.

Happy new year.

Author Response

Reviewer 3: 

The authors have extensively revised and, for the most part, optimized the manuscript according to my comments. Therefore, I find the manuscript publishable. Nonetheless, the quality of all figures (which was asked to be enhanced) is worse than in the first manuscript and should be corrected prior to publication.

Also, just a few comments:

  • Line 167: Please define gene names.
  • Line 194, Table 1: Please delete "Statistics number". Something like "Number of animals", "Number of subjects", etc. would be more appropriate.
  • Figures 1, 2 and 3: It is impossible to read whats in the figures, please fix them.

Response:

Thank you for your constructive comments.

  • Line 167: We defined those gene names as: Follicle Stimulating Hormone Receptor (FSHR), Luteinizing Hormone/Choriogonadotropin Receptor (LHCGR), Estrogen Receptor 1 (ESR1), Pentraxin 3 (PTX3), and BCL2 apoptosis regulator (BCL2)
  • Line 194, Table 1: “Statistics number” was corrected as “Number of animals”.
  • We increased the pixels of Figure 1, 2 , 3 and 4. I'm sorry for my oversight.

Round 3

Reviewer 2 Report

The authors have addressed most of the concerns by the reviewers in the previous revision round and have successfully modified them in this text. It is important to say that the methodology is correct, and the results are very interesting.

English edition have improved the manuscript successfully.

As a minor comment, please add reference(s) to MyM section

My recommendation is to accept the article in its present form